# scCircle-seq unveils the diversity and complexity of extrachromosomal circular DNAs in single cells

Jinxin Phaedo Chen [1,2] ✉, Constantin Diekmann [1,2], Honggui Wu [3,4], Chong Chen [5], Giulia Della Chiara [6], Enrico Berrino [7,8], Konstantinos L. Georgiadis[2,9], Britta A. M. Bouwman[1,2], Mohit Virdi[6], Luuk Harbers [1,2], Sara Erika Bellomo[7], Caterina Marchiò[7,8], Magda Bienko [1,2,6] ✉ & Nicola Crosetto [1,2,6] ✉

Extrachromosomal circular DNAs (eccDNAs) have emerged as important intracellular mobile genetic elements that affect gene copy number and exert in trans regulatory roles within the cell nucleus. Here, we describe scCircle-seq, a method for profiling eccDNAs and unraveling their diversity and complexity in single cells. We implement and validate scCircle-seq in normal and cancer cell lines, demonstrating that most eccDNAs vary largely between cells and are stochastically inherited during cell division, although their genomic landscape is cell type-specific and can be used to accurately cluster cells of the same origin. eccDNAs are preferentially produced from chromatin regions enriched in H3K9me3 and H3K27me3 histone marks and are induced during replication stress conditions. Concomitant sequencing of eccDNAs and RNA from the same cell uncovers the absence of correlation between eccDNA copy number and gene expression levels, except for a few oncogenes, including *MYC*, contained within a large eccDNA in colorectal cancer cells. Lastly, we apply scCircle-seq to one prostate cancer and two breast cancer specimens, revealing cancer-specific eccDNA landscapes and a higher propensity of eccDNAs to form in amplified genomic regions. scCircle-seq is a scalable tool that can be used to dissect the complexity of eccDNAs across different cell and tissue types, and further expands the potential of eccDNAs for cancer diagnostics.

Extrachromosomal circular DNAs (eccDNAs) were originally identified in the 60's as so-called double-minute chromosomes visible in metaphase spreads prepared from childhood leukemia samples[1], and the circular nature of DNA in double-minute chromosomes was subsequently revealed[2–4]. Since then, eccDNAs have been detected in multiple species[5–8] and implicated in various processes, including human tumorigenesis[9–12]. Currently, three main approaches are available to study eccDNAs: (1) DNA fluorescence in situ hybridization (FISH), (2) bulk whole genome sequencing (WGS), and (3) Circle-Seq[13]. DNA FISH has been used to visualize eccDNAs carrying highly

[1]Department of Microbiology, Tumor and Cell Biology, Karolinska Institutet, Stockholm 17177, Sweden. [2]Science for Life Laboratory, Tomtebodavägen 23A, Solna 17165, Sweden. [3]Biomedical Pioneering Innovation Center (BIOPIC), Peking University, Beijing, PR China. [4]School of Life Sciences, Peking University, Beijing, PR China. [5]State Key Laboratory of Biotherapy, West China Hospital, Sichuan University, Chengdu 610041 Sichuan, PR China. [6]Human Technopole, Viale Rita Levi-Montalcini 1, 22157 Milan, Italy. [7]Candiolo Cancer Institute, FPO – IRCCS, Candiolo, SP142, km 3,95, 10060 Turin, Italy. [8]Department of Medical Sciences, University of Turin, Turin, Italy. [9]Department of Oncology and Pathology, Karolinska Institutet, Stockholm 17177, Sweden. ✉e-mail: jinxin.chen@ki.se; magda.bienko@fht.org; nicola.crosetto@fht.org

expressed oncogenes, such as *MYC* and *ERBB2*, which are present in abundant copy numbers in different cancer cell lines[12]. On the other hand, bulk WGS combined with algorithms that try to distinguish between linear genomic DNA (gDNA) and circular DNA has been used to detect eccDNAs in different cancer types[14]. However, this bulk approach can only detect abundant eccDNAs originating from relatively large genomic regions typically characterized by high copy number and structural complexity[9, 11], missing shorter lowly abundant eccDNAs. Circle-Seq[13] identifies eccDNAs by first eliminating linear gDNA via enzymatic digestion, and then enriching circular DNAs through rolling circle amplification (RCA)[15] followed by sequencing. Although Circle-Seq has enabled the identification of a broad spectrum of eccDNAs—including so-called microDNAs and large circular DNAs encompassing well-known oncogenes such as *MYC* (also known as ecDNAs[12])—this method requires millions of cells as input and therefore averages the diversity and complexity of eccDNAs present in a sample. Hence, developing versatile and scalable methods for detecting eccDNAs in single cells or nuclei extracted from tissue biopsies is needed to uncover the full biological complexity of eccDNAs and illuminate their heterogeneity in patient-derived tumor samples. This is especially relevant considering the ability of single-cell sequencing technologies to unveil fundamental aspects of intratumor heterogeneity that cannot be captured by bulk assays, as well as the transformative impact that single-cell technologies are exerting on both basic and translational cancer research[16].

With this goal in mind, here we develop and validate a single-cell adaptation of Circle-Seq – which we named scCircle-seq – that is applicable to both unfixed and fixed cells or cell nuclei, including nuclei extracted from tumor biopsies (Fig. 1a). We first apply scCircle-seq to various cell lines, showing that most eccDNAs are highly variable between cells of the same type, although they tend to preferentially arise from heterochromatic regions enriched in histone 3 methylated on lysine 9 (H3K9me3). Integration of scCircle-seq with the scRNA-seq method Smart-Seq2[17] reveals that the copy number of eccDNAs and the expression of the genes contained within them are typically uncorrelated, except for specific oncogenes, such as *MYC*, that are enclosed within large eccDNAs. We find that the repertoire of eccDNAs of a cell can be used to accurately distinguish between different cell types and that eccDNAs are highly dynamic under replication stress conditions and during cell division. Lastly, we apply scCircle-seq to patient-derived tumor samples representing three different cancer types (luminal B-like and triple-negative breast adenocarcinoma and prostate adenocarcinoma), uncovering tumor-specific eccDNA landscapes and subclonal populations harboring distinct eccDNA genomic patterns. scCircle-seq is an easily scalable, straightforward and versatile method that could potentially be harnessed to unravel the biological complexity and heterogeneity of eccDNAs in cancer.

## Results

### scCircle-seq implementation and technical validation
To develop scCircle-seq, we built on the Circle-Seq protocol[13, 18] by introducing an additional DNA nick repair step to increase the eccDNA detection efficiency, and designed a versatile workflow applicable to both live and fixed cells or nuclei sorted in multi-well plates or single tubes (Fig. 1a and Methods). To analyze scCircle-seq data, we adapted a bioinformatics pipeline previously developed for analyzing bulk Circle-Seq data[11]. Briefly, the pipeline first searches for genomic regions with high sequencing coverage representing putative circle-producing regions (CPRs), and then identifies so-called chimeric junctions inside each CPR by searching over-represented pairs of discordant and split reads (chimeric reads) mapping inside the CPRs (Fig. 1b, Supplementary Fig. 1a, and Methods). A summary of all scCircle-seq experiments is available in Supplementary Data 1. We first tested the specificity of scCircle-seq by manually mixing linear gDNA with (circular) plasmid DNA at

different ratios, which yielded a strong enrichment of reads derived from plasmid DNA and of chimeric junctions joining the extremities of the linear plasmid sequence, as expected (Supplementary Fig. 1b, c). Next, we applied scCircle-seq to five different cell lines (24–49 cells per cell line, 156 cells in total), including four cancer-derived cell lines (HeLa, K562, Colo320DM, and PC3) and one immortalized normal cell line (293T). The number of CPRs identified and the corresponding genome coverage varied between individual cells as well as between cell lines (1.5–8% of the genome depending on the cell line), in agreement with published Circle-Seq data[13] (Fig. 1c, d). Of note, the average length of CPRs was 20 kilobases (kb), with K562 and 293T cells displaying, on average, larger regions (Fig. 1e). The fraction of aligned reads labeled as circular DNA reads was consistently high (70–80%) and using mild lysing conditions or adding a DNA nick repair step significantly increased the number of CPRs detected and the corresponding genome coverage, compared to the conditions used in the bulk Circle-Seq protocol (Supplementary Fig. 1d–h). As a control, we tested a different amplification method (multiple annealing and looping based amplification cycles or MALBAC[19]), which yielded a higher genome coverage but resulted in significantly lower eccDNA enrichment compared to RCA (Supplementary Fig. 1i, j), likely due to amplification of residual linear gDNA in MALBAC. Importantly, the number of chimeric junctions derived from (circular) mitochondrial DNA was comparable between RCA and MALBAC (Supplementary Fig. 1k), indicating that the RCA step in scCircle-seq is not a major source of chimeric read artefacts. However, we cannot completely rule out that some of the chimeric junctions identified by scCircle-seq are false-positive events arising during RCA.

To validate scCircle-seq, we merged single-cell circular DNA profiles from 49 Colo320DM human colon adenocarcinoma cells (pseudo-bulk sample) and compared them to the profile obtained by performing Circle-Seq on ~$10^6$ cells of the same type (Methods). In both cases, the CPRs were enriched in chimeric reads (Supplementary Fig. 2a, b), which are considered as circle-supporting reads in eccDNA detection algorithms[11, 14, 20]. Typically, the CPRs identified in the pseudo-bulk scCircle-seq sample overlapped with those identified by Circle-Seq, although sometimes the patterns differed (Fig. 1f and Supplementary Fig. 2c, d), possibly because of the relatively low number of cells profiled by scCircle-seq ($n = 49$). The borders, coverage, and number of chimeric reads of the CPRs identified by scCircle-seq varied considerably between individual cells (Fig. 1f and Supplementary Fig. 2c, d), indicating that the same larger genomic region can give rise to different eccDNAs in different cells. Visual inspection of chimeric read patterns revealed two main types of CPRs: one is characterized by relatively few chimeric reads mainly connecting the extremities of the region (which we named simple CPRs), while the other contains multiple chimeric reads aligned all along the region (which we named complex CPRs) (Fig. 1g). The latter constitutes the majority (58.3%) of all the CPRs identified in Colo320DM cells and corresponds to regions that give rise to many eccDNAs of high structural complexity. Of note, the number of chimeric junctions increased with the normalized eccDNA copy number (Supplementary Fig. 2e), indicating that genomic regions from which many eccDNAs are produced tend to be associated with a higher complexity of the circles formed, in line with previous findings[21].

### Genomic landscape of eccDNAs in single cells
Next, we sought to investigate the genomic distribution of the eccDNAs detected by scCircle-seq. To this end, we first calculated the autocorrelation of the CPR coverage as a function of genomic distance to determine whether a pattern of eccDNAs exists or whether they randomly form along the genome (Fig. 2a). The autocorrelation rapidly dropped in the case of scCircle-seq data, while the decay was much less pronounced for bulk Circle-Seq (Fig. 2b, c). This suggests that, while

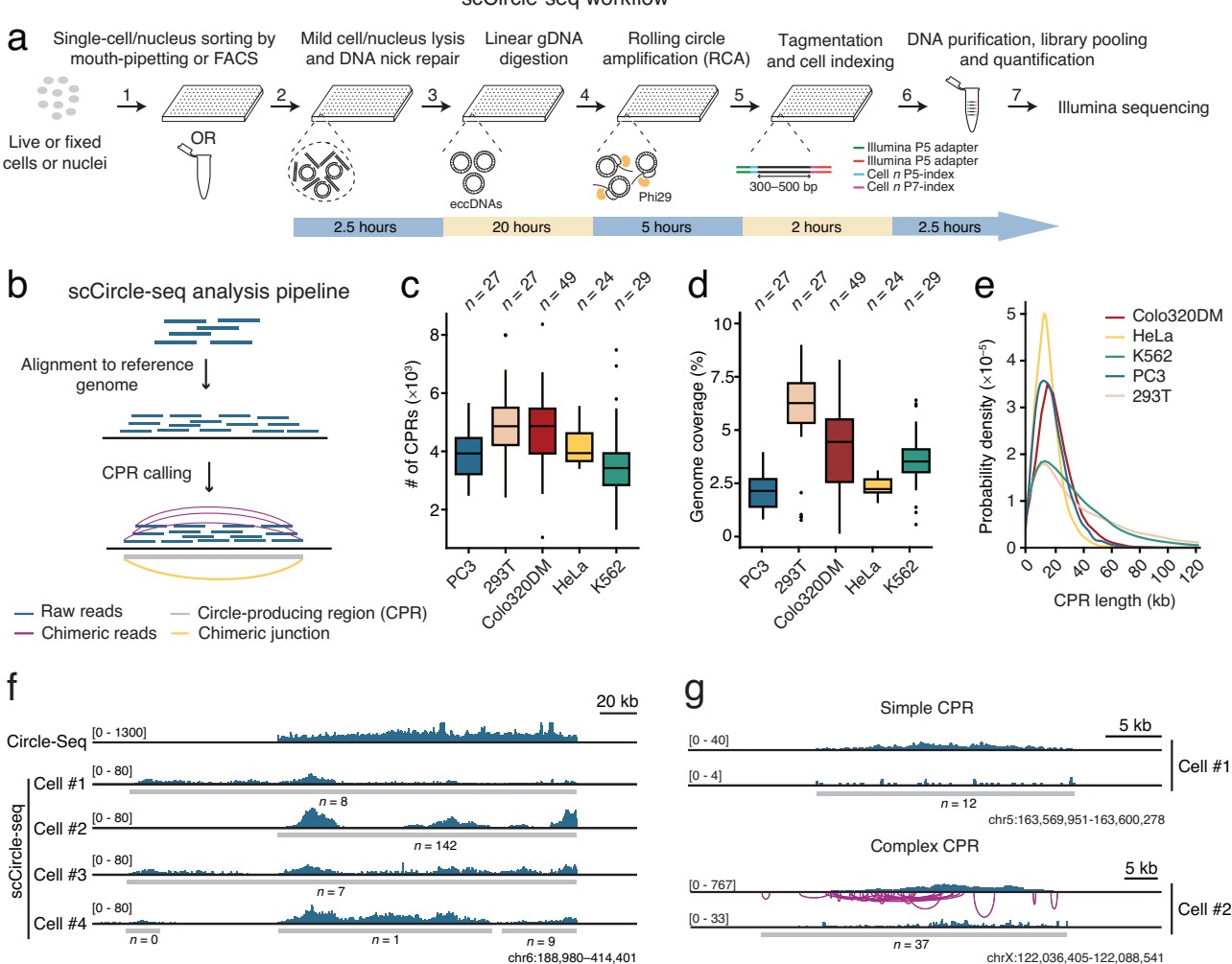

**Fig. 1 | scCircle-seq implementation and validation. a** scCircle-seq workflow. Single nuclei are sorted into 96-well plates or mouth-pipetted into PCR tubes, after which nuclei are lysed and single-stranded DNA breaks (nicks) are repaired using Taq DNA ligase and Bst DNA polymerase. Next, linear genomic DNA (gDNA) is digested with Exonuclease V and the remaining circular DNA is amplified by rolling circle amplification (RCA) using random primers and φ29 DNA Polymerase. The amplified DNA is then subjected to library preparation using the Illumina Nextera kit (see Methods). **b** Scheme of the computational pipeline used to identify circle-producing regions (CPRs). Blue bars, paired-end sequencing reads. Magenta arches, examples of chimeric reads (discordant and split read pairs) defining a chimeric junction (yellow arch) connecting the extremities of a CPR (gray bar). **c** Distributions of the number of CPRs identified by scCircle-seq in five cell lines. *n*, number of single cells analyzed. Boxplots extend from the 25th to the 75th percentile, horizontal bars represent the median, and whiskers extend from −1.5 × IQR to +1.5 × IQR from the closest quartile, where IQR is the inter-quartile range. Black

dots, outliers. In each boxplot, the minimum and maximum are defined, respectively, by the uppermost and lowermost outlier dot or extremity of the corresponding whisker. **d** Same as in (**c**) but for the percentage of the genome covered by CPRs. **e** Probability density distributions of the length of CPRs identified in the five cell lines in (**c**) and (**d**). kb kilobase. **f** Integrative Genomics Viewer (IGV) tracks showing the coverage (dark blue) of the indicated genomic region on chromosome (chr) 6 by Circle-Seq (top track) and scCircle-seq in four different Colo320DM cells. Gray bars, CPRs. **g** Examples of IGV tracks for simple and complex CPRs identified by scCircle-seq on two different chromosomes in Colo320DM cells. For each cell, the upper track indicates the coverage of all reads while the lower track shows the coverage of circle-supporting reads. In (**f**) and (**g**), the numbers in squared brackets represent the intensity range of the track. *n*, number of chimeric junction-supporting reads for the indicated CPRs. Source data are provided as a Source Data file.

eccDNAs are extremely variable at the single-cell level, more defined patterns might be detectable at the population level. To test this possibility, we classified the eccDNAs identified by scCircle-seq in four groups based on (i) the frequency of the corresponding CPRs across all the single cells from the same cell line; and (ii) a so-called uniformity score, which we introduced to measure the overlap of each CPR between individual cells and the corresponding pseudo-bulk sample (Fig. 2d and Methods). In all five cell lines analyzed, the vast majority (88–99%) of eccDNAs detected by scCircle-seq were classified as low frequency low uniformity (LFLU) (Fig. 2e), which correspond to the microDNAs previously identified by Circle-Seq[8]. We also detected high-frequency high uniformity (HFHU) eccDNAs corresponding to the large, oncogene-containing ecDNAs previously identified by Circle-

Seq[9,11] (Fig. 2e). HFHU eccDNAs were mainly detected in the Colo320DM cell line (8.3% of the CPRs) and, to a lesser extent, in K562 (2.5%) and PC3 (0.4%) cells, in agreement with previous observations based on DNA FISH[9,12] (Fig. 2e). High frequency low uniformity (HFLU) eccDNAs were mainly found in Colo320DM (4% of the CPRs) and 293T (3.5%) cells, whereas we detected only a few low frequency low uniformity (LFHU) eccDNAs (Fig. 2e). The coverage and chimeric read patterns of the corresponding CPRs varied considerably between individual cells, even in the case of HFHU eccDNAs (Supplementary Fig. 3a), further highlighting their heterogeneous nature. However, for the two cell lines (Colo320DM and PC3) for which we sequenced sufficient HFHU and LFLU eccDNAs for robust statistics, we found that HFHU CPRs were consistently longer than LFLU CPRs (Supplementary

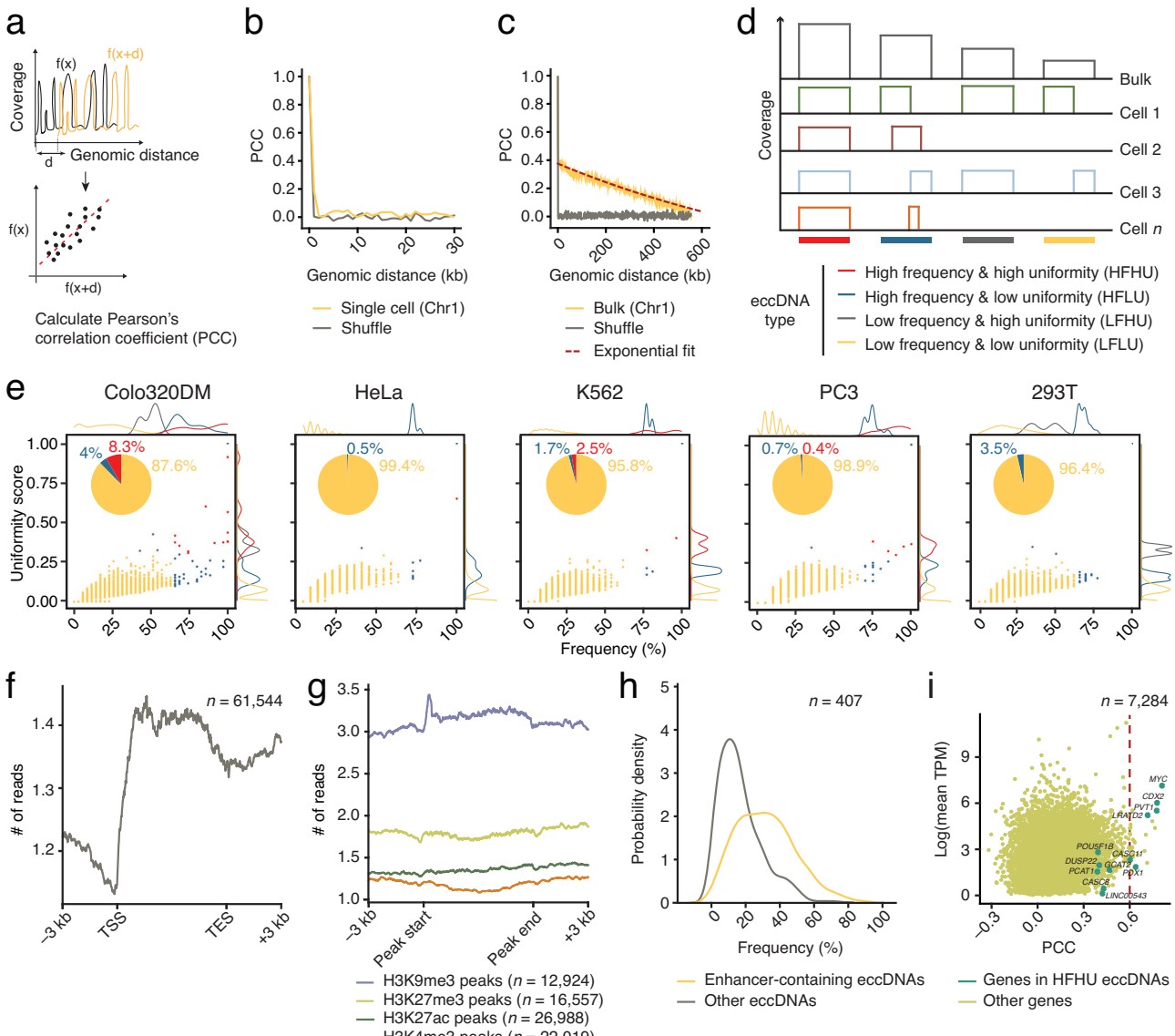

**Fig. 2 | Genomic distribution of eccDNAs identified by scCircle-seq. a** Scheme explaining how we computed the autocorrelation of the scCircle-seq signal as a function of genomic distance. In general, autocorrelation refers to the correlation between a signal and a distance or time-delayed version of itself. Here, auto-correlation refers to the correlation between the scCircle-seq signal $f(x)$ (i.e., the genome coverage of sequencing reads coming from eccDNAs) and the same signal shifted of a genomic distance, $d$ ($f(x+d)$). Autocorrelation of the scCircle-seq signal as function of genomic distance for a single Hela cell (**b**) and for the corresponding pseudo-bulk scCircle-seq dataset from 24 HeLa cells (**c**). Shuffle, auto-correlation after random permutation of the genomic coordinates of scCircle-seq reads. kb, kilobase. **d** Classification of eccDNAs based on their frequency and genome coverage across multiple single cells. **e** Correlation between the frequency and coverage (scatterplots) and relative abundance (pie charts) of the four different types of eccDNAs displayed in (**d**), in each of the five cell types profiled by scCircle-seq. Colors are the same as in (**d**). Each dot in the scatterplots represents a

circle-producing region (CPR). Marginal distributions are shown on the top and right side of each scatterplot. **f** Number of eccDNA reads along the gene body of protein-coding genes in HeLa cells. $n$, number of genes. **g** Number of eccDNA reads inside CPRs overlapping with chromatin immunoprecipitation and sequencing (ChIP-seq) peaks for the indicated histone marks in HeLa cells. $n$, number of ChIP-seq peaks. **h** Probability density distribution of the frequency of CPRs overlapping with enhancers versus all other CPRs in Colo320DM cells. $n$, total number of CPRs. **i** Log-transformed mean transcripts per kilobase million (TPM) versus the Pearson's correlation coefficient (PCC) calculated between the normalized gene expression and normalized number of eccDNA reads for the same gene. Each dot represents a gene. Genes on HFHU eccDNAs are labeled and colored in darker green. The red dashed vertical line serves to visually highlight the minority of genes for which the correlation is relatively strong (PCC > 0.6). $n$, number of genes. Source data are provided as a Source Data file.

Fig. 3b, c). These results indicate that, although eccDNAs are highly heterogeneous in nature, scCircle-seq is able to identify different eccDNA types.

To validate some of the HFHU eccDNAs identified by scCircle-seq with an orthogonal approach, we leveraged our previously described iFISH pipeline[22] to design and produce DNA FISH probes targeting 10 HFHU CPRs identified by scCircle-seq in Colo320DM cells and 3 HFHU CPRs identified in PC3 cells (Supplementary Fig. 4a, Supplementary

Data 2, and Methods). In metaphase spreads, we detected many signals clearly outside of chromosomes (Supplementary Fig. 4b–f), demonstrating the extrachromosomal nature of the eccDNAs detected by scCircle-seq. For some probes (e.g., probes 7–10 in Colo320DM cells), we detected a large number of extrachromosomal signals per cell, whereas for other probes (probes 1–3 in PC3 cells and probes 1–6 in Colo320DM cells) the number of extrachromosomal signals was substantially lower (Supplementary Fig. 4b–f). In the case of Colo320DM

cells, we also detected signals inside metaphase chromosomes (Supplementary Fig. 4d, e), which might reflect the integration of eccDNAs into genomic DNA, as previously reported for this cell line[23].

To further explore the genomic distribution of the eccDNAs detected by scCircle-seq, we intersected the identified CPRs with various genome annotations. We found a sharp depletion of CPRs immediately upstream of the transcription start site (TSS) of protein-coding genes, whereas they were enriched along gene bodies (Fig. 2f). Re-analysis of previously published datasets obtained from different cell types[13] revealed a similar trend of eccDNA depletion near TSSs (Supplementary Fig. 5), corroborating our findings by scCircle-seq. The observed lower density of CPRs around TSSs might be a consequence of the lower DNA bendability previously measured in these nucleosome-depleted regions[24]. To test this hypothesis, we examined whether the local DNA GC-content—which has been associated with DNA bendability[25]—is correlated with the distribution of CPRs along the genome. In all cell lines analyzed except for 293T, the GC-content was significantly lower within the CPRs compared to the rest of the genome, although the difference was minor (Supplementary Fig. 6a–e). This suggests that while DNA bendability might predispose a region to form eccDNAs, this is unlikely a major driver of eccDNA formation. We also did not find a strong correlation between the number of transposon elements in a given CPR and the number of eccDNAs produced from the same region (Supplementary Fig. 6f), contrary to prior observations in normal cells[13, 26]. This could be due to the different mechanisms of eccDNA formation in cancer versus normal cells, as recently proposed[27, 28]. Next, we intersected the CPRs identified by scCircle-seq with chromatin immunoprecipitation and sequencing (ChIP-seq) datasets for various histone modifications available in the Encyclopedia of DNA Elements (ENCODE) (Methods). In all the cell lines analyzed, CPRs were enriched in histone 3 trimethylated on lysine 9 (H3K9me3) marking constitutive heterochromatin and, to a lesser extent, in histone H3K27me3 marking facultative heterochromatin (Fig. 2g and Supplementary Fig. 6g), indicating that eccDNA formation occurs more frequently in heterochromatic regions.

It has been shown that eccDNAs are often associated with enhancer hijacking events in cancer cells[29, 30]. We therefore examined the relationship between the CPRs identified by scCircle-seq and enhancer regions listed in the EnhancerAtlas[31] (Methods). CPRs overlapping enhancer regions were significantly more frequent ($P < 2.2 \times 10^{-16}$, Wilcoxon test, two-sided) than CPRs not overlapping enhancers (Fig. 2h). In Colo320DM cells, CPRs overlapping with enhancers were enriched in sequence motifs recognized by transcription factors previously implicated in colorectal cancer, including MEIS2[32] and ZNF384[33] (Supplementary Fig. 6h), further corroborating the association between eccDNAs and enhancer hijacking.

Previous application of Circle-Seq to profile eccDNAs in Colo320DM cells has revealed the existence of a giant heterotypic ecDNA molecule driving cooperative *MYC* over-expression, which contains multiple enhancers from different chromosomes clustered with a region on chromosome (chr) 8 encompassing the *MYC* oncogene (chr8: 126,424,717–127,997,899)[34]. To test whether scCircle-seq is able to detect the same ecDNA, we computed the co-occurrence between each of the CPRs identified by scCircle-seq and the *MYC*-containing region, across all the 49 Colo320DM cells profiled by scCircle-seq. We identified four CPRs giving rise to HFHU eccDNAs that frequently co-occurred with the *MYC* region (Pearson's correlation coefficient or PCC > 0.5), including one enhancer-containing CPR on chr6 (Supplementary Fig. 6i, j). Notably, heterotypic eccDNAs containing both this enhancer sequence and the *MYC* region on chr8 were detected by DNA FISH in Colo320DM cells (Supplementary Fig. 4e), confirming the existence of the large *MYC*-containing ecDNA previously detected by Circle-Seq and further highlighting the sensitivity of scCircle-seq. We also found three other enhancer-containing CPRs

that frequently co-occurred with the *MYC* region in the same Colo320DM cells profiled by scCircle-seq (Supplementary Fig. 6i, j), suggesting that the ecDNAs that originate from these regions might act as enhancers driving *MYC* overexpression cooperatively. Altogether, these results confirm the previously reported association between eccDNAs and enhancers implicated in oncogene activation and tumorigenesis, further validating scCircle-seq.

## Joint profiling of eccDNAs and mRNAs in the same cell

Next, we examined the relationship between the copy number of the eccDNAs detected by scCircle-seq and the expression of the genes contained within them. To this end, we combined scCircle-seq with the full-length scRNA-seq method Smart-seq2[17], by first separating the cell nucleus from the cytoplasm and then using the former for scCircle-seq and the latter for Smart-seq2 (Methods). We applied this multi-modal approach to three cell lines (Colo320DM, HeLa, and PC3), generating data comparable in both quality and yield (on average, 4,000 CPRs and 5,000 expressed genes detected per cell) to those typically obtained when performing the same assays alone (Supplementary Fig. 7a–d). The correlation between gene expression levels and eccDNA copy number was typically low (PCC < 0.6) for most genes contained within CPRs (7,256 out of 7,284 genes, 99%), even in the case of highly expressed genes (Fig. 2i). Among genes inside CPRs that give rise to HFHU eccDNAs, the PCC was higher than 0.6 only for *CDX2*, *LRAT2*, *MYC*, and *PVT1*, and only in Colo320DM cells (Fig. 2i and Supplementary Fig. 7e–h). These genes are contained within the large ecDNA encompassing the *MYC* locus that was previously detected by Circle-Seq in Colo320DM cells[34] and confirmed by both scCircle-seq and DNA FISH (see previous section). These results are in line with recent findings based on a novel method (scEC&T-seq) for parallel sequencing of ecDNAs and mRNAs in single cells, according to which the copy number of large ecDNAs containing known oncogenes is positively correlated with the expression levels of the same gene in single cells[35]. In contrast, the *MYC*-encompassing large ecDNA found in Colo320DM cells was not detected in PC3 cells, where instead scCircle-seq identified a highly heterogeneous repertoire of shorter eccDNAs for which gene expression levels and eccDNA copy number were uncorrelated (Supplementary Fig. 7i). These results suggest that, even though a subset of oncogene-containing eccDNAs (so-called ecDNAs) that are present in many copies per cell may drive high levels of expression of those oncogenes in certain cell types, the majority of eccDNAs is highly variable from cell to cell and exerts unpredictable effects on the expression of genes contained in them.

## eccDNA based cell type classification

Prompted by these observations, we then wondered whether the eccDNA repertoire of a cell is stochastic or whether cell type-specific eccDNA signatures can be identified using scCircle-seq. To this end, we leveraged cisTopic[36], a computational framework previously designed to analyze scATAC-seq data based on topic modeling, a mathematical approach used in the field of natural language processing[37]. We first represented the genomic distribution of the CPRs detected by scCircle-seq as a *cells × bins* matrix, where *cells* is the number of single cells profiled by scCircle-seq and *bins* are contiguous genomic windows of defined length (Fig. 3a). We then applied cisTopic to decompose the *cells × bins* matrix into two matrices, *cells × topics* and *topics × bins*, and used the decomposed matrices to cluster cells based on topics as well as to annotate each topic using available genomic tracks (Fig. 3a and Methods). In this application, topics can be intuitively thought of as sets of vectors corresponding to different subsets of eccDNAs that can potentially distinguish between different cell types. Indeed, application of cisTopic to our scCircle-seq dataset managed to cluster all the cells belonging to the same cell type together, with the only exception of 2 out of 27 (7.4%) 293T cells that were assigned to the PC3 cell

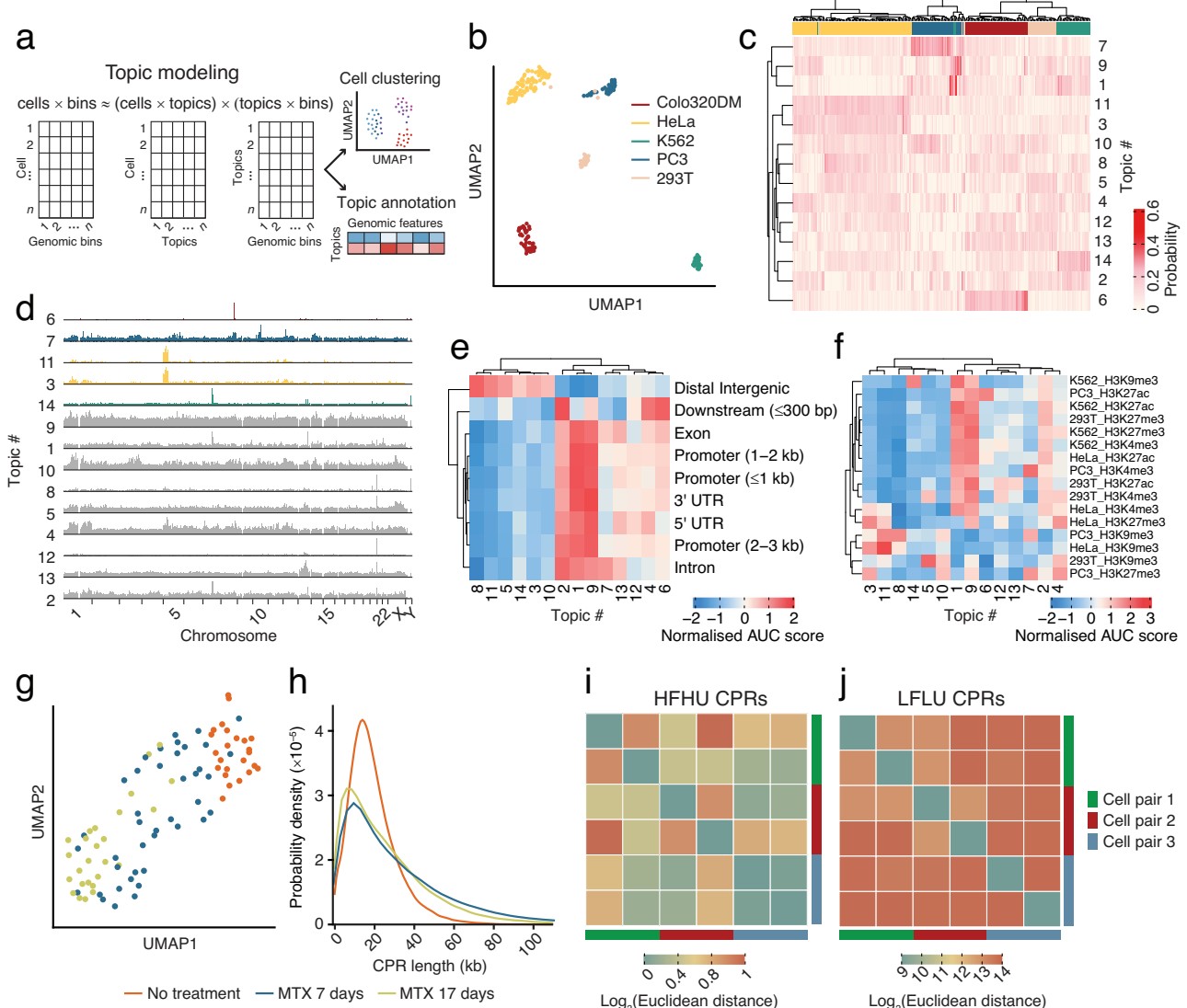

**Fig. 3 | Cell specificity and dynamics of eccDNAs. a** Scheme of the computational approach used for topic modeling of scCircle-seq data using cisTopic[36]. For a detailed description of the approach, see Methods. **b** Uniform Manifold Approximation and Projection (UMAP) representation of scCircle-seq data after topic modeling performed as described in (**a**). Cells are colored by cell type. **c** Heatmap representation of topic contribution for each single cell. Cells and topics are clustered hierarchically. Cells are colored by cell type as in (**b**). **d** Genome-wide contribution to each of the 14 topics identified by topic modeling of scCircle-seq data. Cell type-specific topics are colored by cell type as in (**b**). Gray tracks indicate topics that are not specific to one of the five cell lines processed by scCircle-seq. Heatmap representation of the enrichment in various genomic features (**e**) and histone marks (**f**) for each of the 14 topics identified by topic modeling of scCircle-seq data. Rows and columns are clustered hierarchically. bp base-pair, kb kilobase. **g** UMAP representation of scCircle-seq data from HeLa cells treated or not with methotrexate (MTX). Each dot represents a single cell. **h** Distribution of the length of circle-producing regions (CPRs) identified in the same HeLa cells shown in (**g**). Heatmap representations of the Euclidean distance between pairs of daughter cells for HFHU (**i**) and LFLU (**j**) eccDNAs. Cell pairs are indicated by the colorbars on the right and bottom of each heatmap. Source data are provided as a Source Data file.

cluster (Fig. 3b). 5 out of 14 topics identified by cisTopic were cell type-specific, including one topic specific for Colo320DM cells (topic 6), two for HeLa (topics 3 and 11), one for K562 (topic 14), and one for PC3 (topic 7) (Fig. 3c, d). We did not uncover any topic specific for 293T cells, possibly because this is not a cancer cell line and it might therefore lack a distinctive eccDNA signature. Some topics, including 3 (HeLa specific), 6 (Colo320DM specific), 8, 11 (HeLa specific), 12, and 14 (K562 specific), mark CPRs localized on a few specific genomic regions on one or two chromosomes, whereas the other topics correspond to eccDNAs originating all along the genome (Fig. 3d). Topic 6 (Colo320DM specific) is the most localized topic (Fig. 3d) and corresponds to the region on chr8 that produces the giant ecDNA encompassing the *MYC* oncogene described above. These findings prompted us to assess whether

different topics correspond to specific linear genome features or chromatin types. Indeed, we identified two major topic clusters: one (topics 3, 5, 8, 10, 11, and 14) enriched in distal intergenic elements and the other (topics 1, 2, 4, 6, 7, 9, 12, and 13) enriched in various gene elements (Fig. 3e). Moreover, intersection of the topics with ENCODE ChIP-seq tracks for various histone marks revealed that the cell type-specific topics were strongly enriched in H3K9me3 (topics 3, 11, 14) and H3K4me3 (topic 7) peaks from the corresponding cell types (Fig. 3f, Supplementary Table 1, and Methods). More generally, the topics identified in each cell line were enriched in H3K9me3 and H3K4me3 ChIP-seq peaks from the same cell line (Supplementary Fig. 8a–h). Altogether, these results demonstrate that, although the number and region of origin of eccDNAs widely differ between cells of the same type, different cell types have

specific eccDNA signatures (topics) that partially reflect their epigenome.

## Dynamics of eccDNAs during replication stress and cell division

To further characterize the heterogeneity of eccDNAs and demonstrate the ability of scCircle-seq to capture different eccDNA landscapes, we first treated HeLa cells with the dihydrofolate reductase (DHFR) inhibitor methotrexate (MTX), which has previously been shown to lead to extrachromosomal amplification of the *DHFR* gene driving progressive resistance to MTX[4,38,39]. Exposure of cells to MTX for up to 17 days profoundly rewired the eccDNA landscape of HeLa cells, leading to the formation of longer and more complex circles characterized by a considerably higher fraction of chimeric junctions connecting two different chromosomes or distant regions on the same chromosome (Fig. 3g, h and Supplementary Fig. 9a–c). MTX-induced eccDNAs predominantly originated from constitutive heterochromatin marked by histone H3K9me3 (Supplementary Fig. 9d–h), in line with our finding that CPR topics are enriched in this type of chromatin already in unchallenged cells (see previous section).

To gain further insights into the dynamics of eccDNAs, we applied scCircle-seq to multiple pairs of daughter cells originating from the same Colo320DM parent cell, to examine how eccDNAs are partitioned during cell division (Methods). The pattern of CPRs and chimeric junctions identified by scCircle-seq clearly differed between corresponding daughter cells, even in the case of CPRs giving rise to HFHU eccDNAs (Supplementary Fig. 10a, b). For the latter, the normalized eccDNA copy number typically varied up to 2-fold between daughter cells, whereas in the case of LFLU eccDNAs the difference was much more pronounced, up to 10-fold (Fig. 3i, j). Altogether, these results demonstrate the ability of scCircle-seq to portray the dynamics of eccDNAs, and reveal that the eccDNA repertoire of a cell rapidly diverges after mitosis, in line with recent observations based on live imaging[40,41].

## scCircle-seq on patient-derived tumor samples

Lastly, we sought to demonstrate the applicability of scCircle-seq to patient-derived tumor samples, which is essential to gain insights into the heterogeneity and clinical implications of eccDNAs across the spectrum of human cancers. As a proof-of-concept, we applied scCircle-seq to nuclei extracted from three retrospectively collected frozen tumor samples and individually sorted in 96-well plates using fluorescent activated cell sorting (FACS) (one prostate adenocarcinoma, PRAD; one triple-negative breast cancer, TNBC; one luminal B-like breast cancer, LumB) (Methods). We obtained high-quality sequencing data for 55 PRAD, 87 TNBC and 33 LumB nuclei, achieving nearly 100% circular-to-linear spike-in DNA ratio in all the cells (range: 99.5–99.9%) and 50–80% of all the reads assigned to eccDNAs (Supplementary Fig. 11a, b). The vast majority of the eccDNAs identified in these tumor samples was classified as low frequency low uniformity (LFLU) (Supplementary Fig 11c). These results are consistent with those obtained from immortal cell lines (Supplementary Fig. 1b, d), demonstrating that scCircle-seq performs robustly even on nuclei extracted from tumor biopsies. The largest number of CPRs was identified in PRAD cells, followed by TNBC and LumB (Fig. 4a). Similarly, the fraction of the genome covered by CPRs was the highest for PRAD, followed by TNBC and LumB cells (Fig. 4b), suggesting that these tumors harbor different eccDNA landscapes. Indeed, dimensionality reduction of single-cell CPR genomic profiles using Uniform Manifold Approximation and Projection (UMAP)[42] revealed two distinct clusters clearly separating PRAD from TNBC and LumB cells (Fig. 4c). Differential topic analysis identified multiple topics enriched either in TNBC and LumB cells or in PRAD cells (Fig. 4d and Supplementary Fig. 11d), indicating that these two groups of tumors carry different eccDNA repertoires, in line with their different tissue of origin.

We then wondered whether, despite originating from the same tissue, also TNBC and LumB cells have distinct eccDNA landscapes, which might reflect the different biology and clinical behavior of the corresponding tumors. UMAP analysis identified two major clusters (1 and 2) comprising both TNBC and LumB cells and harboring distinct CPR landscapes, as well as two minor clusters containing only LumB cells surrounded by cells assigned to Cluster 1 in the 2D UMAP representation (Fig. 4e, f and Supplementary Fig. 11e). Differential topic analysis pinpointed multiple topics specific to either Cluster-1 TNBC or Cluster-2 TNBC cells (Fig. 4g and Supplementary Fig. 11f), suggesting that these two groups might represent two different cell populations with distinct mechanisms of eccDNA formation within the same tumor.

We hypothesized that the two major TNBC clusters identified might correspond to cells harboring different levels of chromosomal instability, which in turn might result in different eccDNA repertoires. To test this hypothesis, we first intersected the CPRs identified in LumB cells and TNBC cells assigned to either Cluster 1 or 2 with somatic copy number alterations (SCNAs) identified in breast cancers sequenced as part of The Cancer Genome Atlas (TCGA)[43]. We found that amplified regions were significantly enriched in the CPRs of Cluster 1 TNBC and LumB cells compared to the CPRs of TNBC Cluster 2 cells (Fig. 4h, i). Next, we performed single-cell DNA sequencing by Acoustic Cell Tagmentation (ACT)[44] on 384 nuclei extracted from the same TNBC sample to assess the relationship between eccDNAs and SCNAs in the same sample (Supplementary Data 3 and Methods). Phylogenetic analysis of 174 high-quality single-cell copy number profiles revealed two major groups of cells: one group predominantly composed of diploid cells or cells with sparse copy number alterations, and another group composed of cells with multiple SCNAs, representing bona fide tumor cells (Supplementary Fig. 12). As for TCGA SCNAs, amplified regions identified by ACT were significantly enriched in TNBC Cluster-1 CPRs (Fig. 4j, k). Lastly, we examined whether the number of eccDNAs identified scales with the copy number of a given genomic region. We found that the number of CPRs did not significantly differ between diploid and moderately amplified (3–4 copies) regions, whereas they were significantly more abundant in regions with higher amplification levels (5–6 copies) (Fig. 4l). Altogether, these results reveal that the landscape of eccDNAs of tumor cells reflects their SCNA landscape, and that highly amplified genomic regions are associated with a higher number of eccDNAs produced.

## Discussion

We have developed a single-cell version of Circle-Seq[18] that allows interrogating the diversity and complexity of eccDNAs at the single-cell level, including in nuclei extracted from cryopreserved tumor biopsies. scCircle-seq leverages the power of rolling circle amplification (RCA) and a DNA nick repair step to achieve high detection sensitivity, enabling the detection not only of abundant, oncogene-encompassing ecDNAs, but also of rare eccDNAs. Recently, another method for single-cell eccDNA profiling (scEC&T-seq) was developed and applied to investigate the structural and functional heterogeneity of eccDNAs in neuroblastoma cell lines and primary tumor samples[35]. While this represents a significant technological advancement, the published scEC&T-seq protocol is difficult to scale up as it entails multiple DNA purification steps before the eccDNAs from each single cell are amplified by RCA and indexed prior to sequencing. This can lead to loss or breakage of eccDNAs during the procedure, while being significantly more time-consuming compared to scCircle-seq (5 days of linear gDNA digestion plus 20 h of RCA in scEC&T-seq, compared to 24 h of digestion and 5 h of RCA in scCircle-seq, see Fig. 1a). Importantly, in scCircle-seq there is no intermediate DNA purification before RCA and single-cell indexing, which considerably simplifies the workflow and reduces the risk of losing low-abundant eccDNAs.

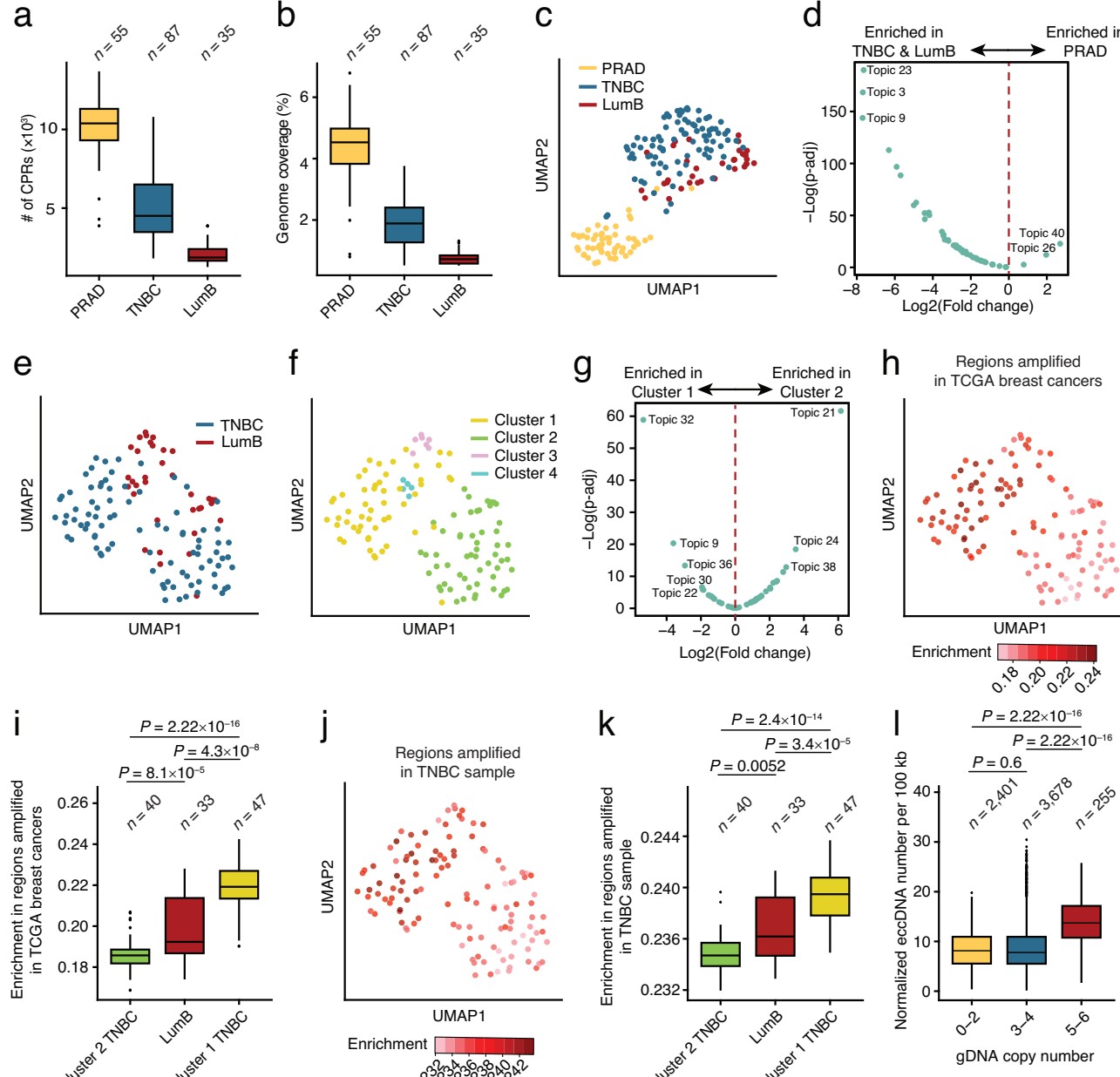

**Fig. 4 | scCircle-seq detects eccDNAs in patient-derived tumor samples.**
**a** Distributions of the number of circle-producing regions (CPRs) in nuclei extracted from three tumor samples. PRAD, prostate adenocarcinoma. LumB, Luminal B-like breast cancer. TNBC, triple-negative breast cancer. *n*, number of cells analyzed. **b** As (**a**) but for the genome coverage of the CPRs identified. **c** Uniform Manifold Approximation and Projection (UMAP) representation of all cells analyzed, after topic modeling of the corresponding eccDNAs. Each dot represents one cell. **d** Volcano plot showing differentially expressed topics. −Log(p-adj), negative logarithm of the adjusted *P* value calculated using the Benjamini−Hochberg method (two-sided, pair-wise). Log2(fold change), base-2 logarithm of the fold change. **e, f** UMAP representation of LumB and TNBC cells, after topic modeling of the corresponding eccDNAs. Left, Cells colored by sample type. Right, Cells colored by clusters identified by unsupervised clustering. **g** As (**d**) but comparing Cluster-1 and Cluster-2 from (**f**). **h** As (**e**) but with nuclei color-coded based on enrichment of the corresponding eccDNAs inside genomic regions amplified in breast cancer

samples in The Cancer Genome Atlas (TCGA)[43]. **i** Distributions of the enrichment inside genomic regions amplified in TCGA breast cancers, for the eccDNAs identified in LumB and TNBC cells belonging to UMAP Cluster-1 and Cluster-2 in (**f**). *n*, number of single cells analyzed. *P*, *t*-test, two-tailed. **j, k** Same as in (**h**) and (**i**), respectively, but for genomic regions amplified in the TNBC sample based on single-cell DNA sequencing using Acoustic Cell Tagmentation (ACT)[44].
**l** Distributions of the normalized number of eccDNAs per 100 kilobases (kb) inside genomic regions with different copy numbers determined by ACT. *P*, *t*-test, two-sided. Genomic regions are grouped based on the corresponding copy number. In all the boxplots, boxes extend from the 25th to the 75th percentile, horizontal bars represent the median, and whiskers extend from −1.5 × IQR to +1.5 × IQR from the closest quartile, where IQR is the inter-quartile range. Black dots, outliers. Minimum and maximum are defined, respectively, by the uppermost outlier dot or extremity of the corresponding whisker. Source data are provided as a Source Data file.

A major advantage of scCircle-seq compared to bulk Circle-Seq is that it enables detecting low frequency circular DNAs that are only present in a small fraction of the cells, but that, collectively, constitute the vast majority of the eccDNA repertoire. By profiling eccDNAs across five different cell lines, here we have shown that the vast majority of eccDNAs detected can be classified as low frequency low uniformity (LFLU) events that widely differ between individual cells, both in terms of their genomic location and in their complexity.

Although we cannot completely rule out the possibility that some chimeric junctions are artefacts of RCA, our data indicate that multiple eccDNAs can arise from the same genomic region in a cell. Multiple eccDNA molecules originating from the same region might also fuse among themselves or with circles originating from other regions, in line with prior observations[21], contributing to the complexity of the single-cell eccDNA landscape. Of note, LFLU eccDNAs were also recently described by others under the name of stochastic eccDNAs[35], corroborating our findings. However, the validation of such low-frequency eccDNAs with orthogonal approaches such as FISH remains technically challenging, because of the high heterogeneity in the length and sequence of the eccDNA molecules that are produced from the same genomic region across different cells. We anticipate that emerging in situ sequencing approaches, such as in situ genome sequencing (IGS)[45], might be harnessed in the future to orthogonally validate LFLU eccDNAs detected by single-cell sequencing approaches.

Although the eccDNA landscapes uncovered by scCircle-seq mainly consist of low frequency and structurally diverse eccDNAs, we have found that certain features of these eccDNAs (i.e., the topics identified by cisTopic[36]) are cell type-specific and are strongly related to the distribution of H3K9me3 (constitutive heterochromatin) and H3K4me3 (active enhancer) histone marks in the same cells. This indicates that the epigenomic landscape of a cell, which contributes to define its identity, influences where and which type of eccDNAs are produced along the genome. Notably, our observation that cell treatment with methotrexate—which causes replication stress and consequently DNA breaks—results in the formation of eccDNAs with higher structural complexity preferentially inside constitutively heterochromatic regions marked by histone H3K9me3, suggests that increased formation and/or slower repair of DNA breaks in these regions might facilitate the formation of eccDNAs.

In addition to applying scCircle-seq to cultured cell lines, we have demonstrated that our method can also be applied to nuclei extracted from cryopreserved tumor biopsies and sorted in multi-well plates. By profiling the eccDNAs in single nuclei from three different tumor types, we have shown that different tumors harbor distinct eccDNA landscapes that, at least in part, correlate with the DNA copy number profile of the corresponding tumors. Indeed, in a triple-negative breast cancer sample, which we profiled by both scCircle-seq and single-cell DNA-seq, we found two clearly distinct cell subpopulations, one of which was enriched in eccDNAs originating from amplified genomic regions identified in the same tumor. Notably, the circle-producing regions identified in the same subpopulation were also over-represented inside genomic regions frequently amplified across thousands of breast cancers sequenced in TCGA. Thus, scCircle-seq can be used to dissect the heterogeneity of eccDNAs in tumor samples and pinpoint different subpopulations of cells carrying distinct eccDNA repertoires that likely reflect different underlying genome instability processes.

In conclusion, we have developed a straightforward, sensitive, and versatile method for dissecting the heterogeneity and structural complexity of eccDNAs in both cell and tissue samples, including tumor biopsies. scCircle-seq can be used to explore the diversity of eccDNAs in different cell and tissue types, contributing to understanding the origin and functional implications of this fascinating form of genetic information transfer. We anticipate that future applications of scCircle-seq to large cohorts of tumor samples will further expand our knowledge on the heterogeneity of cancer genomes and pave the way to the use of eccDNAs in cancer diagnostics.

## Methods
### Ethical statement
The work described here complies with all ethical regulations relevant to the committees approving the two respective permits (Södersjukhuset hospital in Stockholm, Sweden for #2018/1003-31 and Candiolo Cancer Institute FPO - IRCCS, Turin, Italy for 001-IRCC-00IIS-10). The patient-derived tumor samples used in this study have been collected and used in conformity to the permits. All donors analyzed were of European ethnicity. None of the donors received compensation and written informed consent was obtained for all donors involved in both cohorts. During clinical evaluation, the seven donors available in the prostate cancer cohort reported male sex and ages ranging from 43–65 years old. One sample was excluded due to low sample quality, and one of the six remaining samples was used here. For the breast cancer samples, both donors included here reported female sex and they were 58 and 65 years of age at the time of sample collection.

### Experimental methods
#### Samples
**Cell lines.** We obtained all the cell lines used in the study from ATCC: Colo320DM (cat. no. CCL-220), PCR3 (cat. no. CRL-1435), HeLa (cat. no. CCL-2), HEK293T (cat. no. CRL-1573), K562 (cat. no. CCL-243). We authenticated all cell lines by STR genotyping. We cultured Colo320DM and PC3 cells in RPMI-1640 medium (Gibco, cat. no. C11875500BT) supplemented with 15% fetal bovine serum (Gibco, cat. no. 10091148) and 1% penicillin–streptomycin (Gibco, cat. no. 15140122); HeLa and HEK293T cells in DMEM medium (Gibco, cat. no. C11995500BT) supplemented with 10% fetal bovine serum; and K562 cells in IMDM medium (Gibco, cat. no. C12440500BT) supplemented with 15% fetal bovine serum. Colo320DM and K562 cells were cultured at 37 °C with 10% $CO_2$ while PC3, HeLa, and HEK293T cells were cultured at 37 °C with 5% $CO_2$.

**Patient-derived tumor samples.** For scCircle-seq on prostate cancer, we extracted nuclei from one frozen tissue block excised from one of six prostatectomy samples that we previously collected at the Södersjukhuset hospital in Stockholm, Sweden for single-cell spatially resolved profiling DNA copy number alterations (ethical permit #2018/1003-31). For scCircle-seq on breast cancer, we extracted nuclei from two frozen breast cancer specimens (one classified as Luminal B-like [herein labeled LumB] and the other as triple-negative [herein labeled TNBC]) previously collected and stored at the Pathology Unit of the Candiolo Cancer Institute FPO - IRCCS, Turin, Italy (ethical permit "Profiling", 001-IRCC-00IIS-10). All the patient-derived samples described in this study are unique biological samples that cannot be distributed to other researchers.

**scCircle-seq.** A step-by-step scCircle-seq protocol is available in Protocol Exchange at the following https://protocolexchange. researchsquare.com/article/pex-2385/v1.

**scCircle-seq on cell lines.** We mouth pipetted single cells into PCR tubes containing 0.25 μL of Dynabeads MyOne Silane beads (Invitrogen, cat. no. 37002D) diluted in 6.75 μL of nucleus isolation buffer containing 10 mM Tris-HCl pH 7.5, 10 mM NaCl, 3 mM $MgCl_2$, 0.1% Tween-20 (Invitrogen, cat. no. 003005), IGEPAL CA-630 0.3% (Sigma-Aldrich, cat. no. 18896), 0.1% bovine serum albumin (Sigma-Aldrich, cat. no. A2934), and 2 mM dithiothreitol. After incubation on ice for 30 min we gently vortexed the tubes for 1 min, followed by centrifugation at 500 × $g$ for 5 min at 4 °C. Afterwards, we transferred 5.4 μL of supernatant containing cytoplasmic RNA into a new 0.2 mL tube for scRNA-seq using the Smart-seq2 library preparation approach (see below), leaving the bead pellet containing the nuclei undisturbed. After nucleus isolation, we added 0.4 μL of NEBNext FFPE DNA Repair Mix (New England Biolabs, cat. no. M6630S) containing 0.25 ng/mL linear spike-in DNA, 0.25 ng/mL circular spike-in DNA, 5X NEBNext FFPE DNA Repair Buffer, and 0.1225X NEBNext FFPE DNA Repair Mix into each tube containing a bead pellet, gently vortexed the samples,

and incubated them at 20 °C for 1 h. After nick repair, we added 1.54 μL of nuclear lysis mix containing 40 mM Tris-HCl, 40 mM NaCl, 0.2% TritonX-100 (Sigma-Aldrich, cat. no. T9284), 30 mM dithiothreitol, 2 mM EDTA, and 1.6 μg/μL Qiagen Protease (Qiagen, cat. no. 19157) into each tube, gently vortexed the samples, and incubated them at 50 °C for 30 min followed by holding at 4 °C. Next, we added 0.05 μL of Protease Inhibitor cocktail (Sigma-Aldrich, cat. no. 8340) and 0.45 μL of water to each tube and incubated the samples at 37 °C for 1 h. After nuclear lysis, we performed linear DNA digestion by adding 1.2 μL of digestion mix containing 8.3 mM ATP, 4.16X Plasmid-Safe Reaction Buffer (Lucigen, cat. no. E3101K), 0.83–2.49 U/μL Plasmid-Safe ATP-Dependent DNase (Lucigen, cat. no. E3101K) depending on the ploidy of the cells (for diploid cells: 0.83 U/μL, for tetraploid: 2.49 U/μL), and 2.08 mM dithiothreitol into each tube, and incubated the samples at 37 °C for 20 h, followed by 70 °C for 10 min and holding at 4 °C. Next, we added 5 μL of amplification mix containing 2X phi29 buffer (New England Biolabs, cat. no. M0269), 2 mM dNTPs (Thermo Fisher Scientific, cat. no. R0192), 100 μM Exo-Resistant Random Primer (Thermo Fisher Scientific, cat. no. SO181), 0.002 U/μL Pyrophosphatase inorganic (Therm-Fisher Scientific, cat. no. EF0221), and 1.6 U/μL Phi29 DNA Polymerase (New England Biolabs, cat. no. M0269) into each tube, and incubated the samples at 30 °C for 2 h followed by 65 °C for 10 min and holding at 4 °C. We purified the amplified circular DNA on DNA Clean & Concentrator-5 columns (Zymo Research, cat. no. D4014), after which we used 10 ng of purified circular DNA as input for the Nextera XT DNA Library Preparation Kit (Illumina, cat. no. FC-131-1024).

**scCircle-seq on nuclei from tumor biopsies.** We first isolated nuclei from the prostate cancer (PRAD), luminal B (LumB) and triple-negative breast cancer (TNBC) samples described above, using an ad hoc modified version of the Nuclei extraction from frozen tissue for single-nuclei sequencing protocol from Mission Bio (https://support.missionbio.com/hc/en-us/articles/360042902014-Nuclei-Extraction-From-Frozen-Tissue-User-Guide). Briefly, we first prepared a tissue lysis solution (TLS) containing 0.03 mg/mL Trypsin-EDTA (0.25%), phenol red (Thermo Fischer Scientific, cat. no. 25200072), 0.1 mg/mL Collagenase type7 (Worthington, cat. no. CLS-7 LS005332) and 0.1 mg/mL Dispase II (Gibco, cat. no. 17105-041) in a spermine solution (pH 7.6) containing 3.4 mM sodium citrate tribasic dihydrate (Sigma-Aldrich, cat. no. C8532), 1.5 mM spermine tetrahydrochloride (Sigma-Aldrich, cat. no. S1141), 0.5 mM tris (hydroxymethyl) aminomethane (Sigma-Aldrich, cat. no. 252859), and 0.1% v/v IGEPAL CA-630 (Sigma-Aldrich, cat. no. I8896) in molecular biology grade water. We then added 200 μl of ice-cold TLS onto each tissue block kept into a pre-chilled Petri dish on dry ice and incubated the samples until the TLS had frozen (~3 min). After initial mincing on dry ice using a pair of pre-chilled sterile scalpels, we transferred the tissue to room temperature and continued mincing until the tissue was dissociated into small fragments that could flow through a 1 mL pipette tip. We then added 1.8 mL of TLS and transferred the whole volume of TLS solution with the tissue fragments into a 5 mL low-binding tube (Sigma-Aldrich, cat. no. EP0030108310-200EA). We incubated the samples for 15 min at room temperature on a device rotating at 20 rpm, after which we added 2 mL per sample of a stop solution containing 25 mg of Trypsin inhibitor from chicken egg white, Type II-O (Sigma-Aldrich, cat. no. T9253), and 5 mg Ribonuclease A from bovine pancreas, Type I-A (Sigma-Aldrich, cat. no. R4875) dissolved in 49.8 mL of spermine solution. We gently inverted each tube 15 times, after which we filtered the tissue suspension through a 50 μm CellTrics cell strainer (Sysmex, cat. no. 04-004-2327) and centrifuged the flowthrough at $300 \times g$ for 5 min at room temperature. We discarded the supernatant, resuspended the pellet containing the nuclei in 400 μL of nuclei fixation solution (66% Methanol, 33% Acetic acid) and incubated the samples on ice for 15 min. Following centrifugation at $300 \times g$ for

5 min at room temperature, we discarded the supernatant, resuspended the nuclei pellet in 1 mL of 1X PBS with 5 mM EDTA, and filtered the nuclei suspension again through a 10 μm CellTrics cell strainer (Sysmex, cat. no. 04-004-2324). We stored the nuclei suspension at 4 °C until sorting.

For single-nucleus sorting, we first added propidium iodide (PI, Thermo Fisher Scientific, cat. no. P3566) to the fixed nuclei suspensions to reach 1 mg/mL final concentration. We sorted PI+ nuclei into low DNA binding 96-well plates (Eppendorf, cat. no. 0030129504) pre-filled with 1.4 mL/well of nucleus isolation buffer, using the MoFlo Astrios EQs (Beckman Coulter) FACS system, excluding the doublets. Immediately after sorting, we quickly centrifuged the nuclei at $300 \times g$ for 3 min at +4 °C and then stored the plates at −20 °C before proceeding to scCircle-seq using the same procedure as described above for cell lines.

For preparing libraries, we diluted the RCA products 20 times with nuclease-free water and transferred 1 μL of each RCA product (corresponding to one nucleus) into a new 96-well plate as input for tagmentation using the Nextera XT DNA Library Preparation Kit (Illumina, cat. no. FC-131-1024).

**Sequencing.** For scCircle-seq on cultured cells, we pooled all the libraries in paired-end mode on a HiSeq X Ten (Illumina) machine. For scCircle-seq on prostate cancer nuclei, we sequenced all the cells in paired-end mode on a NextSeq 2000 (Illumina) machine using the NextSeq 1000/2000 P2 Reagents (300 Cycles) v3 (Illumina, cat. no. 20046813). For scCircle-seq on breast cancer nuclei, we pooled the LumB and TNBC samples and sequenced them in paired-end mode on a NovaSeq 6000 (Illumina) machine using the NovaSeq 6000 SP Reagent Kit v1.5 (300 cycles) (Illumina, cat. no. 20028400).

**scCircle-seq in daughter cells.** To study how eccDNAs are passed to daughter cells during mitosis, we cultured Colo320DM cells in a 10 cm Petri dish at low density (1000 cells per plate) to ensure that each cell is separated from its neighbors. After most cells underwent one mitosis (~10 h), we gently discarded the medium, added 4 mL trypsin (Gibco, cat. no. 25200056) onto the cells and incubated for 1 min at room temperature. Using a mouth pipette, we isolated several pairs of daughter cells and placed each daughter cell into a separate tube for scCircle-seq.

**Bulk Circle-Seq.** We performed bulk Circle-Seq according to the previously described protocol[7, 10], including the following modifications. In brief, we harvested 1 million cells during the exponential growth phase and extracted high molecular weight genomic DNA with the MagAttract HMW DNA Kit (Qiagen, cat. no. 67563). Next, we digested 1 μg of DNA in 100 μL of digestion mix containing 20 U of Plasmid-Safe ATP-Dependent DNase (Lucigen, cat. no. E3101K), 25 mM ATP (Lucigen, cat. no. E3101K), 1X Plasmid-Safe Reaction Buffer (Lucigen, cat. no. E3101K) for 6 days at 37 °C. Every 24 h we replenished the enzymes in the digestion mix by adding 2 μL of fresh Plasmid-Safe ATP-Dependent DNase, 4 μL of ATP, and 0.6 μL of Plasmid-Safe 10X Reaction Buffer. After 6 days, we purified circular DNA with 1X AMPure XP beads (Beckman Coulter, cat. no. A63881) following the manufacturer's instructions. Lastly, we used 20 ng of purified circular DNA as input for the Nextera XT DNA Library Preparation Kit (Illumina, cat. no. FC-131-1024) and sequenced the libraries on a HiSeq X Ten (Illumina) machine, aiming at generating around 10 million reads per sample.

**Smart-seq2.** We performed Smart-seq2 following the previously published protocol[17]. Briefly, we mixed 5.4 μL of supernatant containing cytoplasmic RNA obtained from the nucleus isolation step in scCircle-seq with 1.27 μL of oligo-dT mix containing 1 μM oligo-dT30VN (5′−AAGCAGTGGTATCAACGCAGAGTACT30VN-3′) and 1 mM (each) dNTPs and incubated the mix for 5 min at 72 °C followed by 5 min on

ice. Next, we added 7.2 μL of a reverse transcription mix containing 10 U/μL SuperScript II reverse transcriptase (Invitrogen, cat. no. 18064071), 1 U/μL SUPERase (Invitrogen, cat. no. AM2696), 1X Superscript II first-strand buffer (Invitrogen, cat. no. 18064071), 1 mM GTP, 5 mM dithiothreitol, 1 M betaine (Sigma-Aldrich, cat. no. B0300), 1 mM MgCl₂, and 1 μM template-switching oligo (5′-AAGCAGTGGTAT-CAACGCAGAGTACATrGrG+G-3′) to each sample and incubated the samples for 90 min at 42 °C, followed by 11 cycles of: 50 °C for 2 min, 42 °C for 2 min, 70 °C for 5 min. Lastly, we added 14.52 μL of amplification mix containing 1X KAPA HiFi HotStart ReadyMix (Roche, cat. no. KK2602) and 0.1 μM ISPCR oligo (5′-AAGCAGTGGTATCAACGCAGAGT-3′) to each sample and performed PCR with the following settings: 98 °C for 3 min; 21 cycles of: 98 °C for 20 s, 65 °C for 30 s, 72 °C for 4 min; 72 °C for 15 min; 4 °C on hold. We purchased all the primers from IDT as standard desalted primers.

### Induction of eccDNAs with methotrexate

To induce the production of eccDNAs, we used an approach based on cell treatment with the chemotherapeutic agent, methotrexate, as previously described[39]. Briefly, we grew HeLa cells into 6-well plates containing medium supplemented with 100 nM methotrexate (Sigma-Aldrich, cat. no. 1414003), replacing the medium with fresh one every 2 days. We mouth pipetted single cells for scCircle-seq at day 7 and 17.

### DNA fluorescence in situ hybridization (FISH)

To demonstrate the extra-chromosomal nature of eccDNAs, we performed DNA FISH on metaphase spreads of Colo320DM, K562, and PC3 cells, targeting some of the high frequency high uniformity (HFHU) eccDNAs identified by scCircle-seq in these cells (see Supplementary Fig. 4). To prepare metaphase spreads, we grew Colo320DM, K562 and PC3 cells for 24 hours in their culture medium supplemented with colcemid (KaryoMax, Thermo Fisher Scientific, cat. no. 15210040) at a concentration of 2 μg/mL, 0.2 μg/mL and 0.1 μg/mL, respectively. Afterwards, we collected the cells, permeabilized them with 0.075 M KCl hypotonic solution for 15 min at 37 °C and fixed them with Carnoy's fixative (methanol:acetic acid 3:1, v/v) for 10 min at room temperature. To obtain metaphase spreads, we gently dropped the fixed cells onto cold, pre-humidified coverslips from a height of ~1 m above, and left the coverslips air-dry. We designed and produced oligonucleotide DNA FISH probes using our previously described iFISH pipeline[22]. The sequences of all the oligos composing the probes used are available in Supplementary Data 2. To perform DNA FISH, we followed the step-by-step protocol for oligo-based DNA FISH that we previously described[22]. We imaged the samples on ×100 1.45 NA objective mounted on a custom-built Eclipse Ti-E inverted microscope system (Nikon) controlled by the NIS Elements software (Nikon) and equipped with an iXON Ultra 888 ECCD camera (Andor Technology), selecting 6−10 fields of view (FOVs) per sample containing metaphase spreads. In these FOVs, eccDNAs appear as individual fluorescence spots clearly separated from metaphase chromosomes and interphase nuclei, and not overlapping with the DNA signal.

### Single-cell DNA-seq by Acoustic Cell Tagmentation (ACT)

To study the relationship between eccDNA production and DNA copy number alterations, we adapted the protocol for Acoustic Cell Tagmentation (ACT)[44] on a non-acoustic based nanodispensing device (I.DOT, Dispendix GmbH). Briefly, we FACS-sorted single nuclei in 384-well plates prefilled with 5 μL of Vapor-Lock (Qiagen, cat. no. 981611) per well. For cell lysis, we lysed each nucleus in 150 nL of lysis buffer containing 20 mM Tris pH8, 20 mM NaCl, 25 mM DTT, 0.15% Triton X-100, 1 mM EDTA, and 25 μg/mL Qiagen Protease (Qiagen, cat. no. 19157). After dispensing, we centrifuged the plate at 3000 × g for 3 min, vortexed it at 1000 rpm for 1 min, and then again centrifuged it at 3220 × g for 3 min. This was done after every dispensing step with

I.DOT. For lysis, we incubated the plate at 50 °C for 1 h followed by heat inactivation at 70 °C for 15 min. To neutralize EDTA in the lysis buffer, we dispensed 50 nL of 4 mM MgCl₂ into each well and then vortexed and centrifuged the plate. For tagmentation, we dispensed 600 nL of tagmentation reaction mix containing Tagmentation DNA buffer (TD) and Amplicon Tagment Mix (ATM) at 2:1 v/v ratio (Nextera kit, Illumina, cat. no. FC-131-1096) into each well and performed tagmentation at 55 °C for 5 min followed by hold at 4 °C in a PCR thermocycler. To stop the reaction, we dispensed 200 nL of neutralization (NT) buffer into each well and incubated the plate for 5 min at room temperature. Lastly, we performed single-nucleus indexing by dispensing 1.35 μL of PCR master mix containing 1.3 μL of 2X Q5 Master Mix (New England Biolabs, cat. no. M0492L) and 50 nL of 100 mM MgCl₂, and then 100 nL each of P5 and P7 Nextera index primers (Illumina, cat. no. 20027213, 20027214, 20042666, 20042667) into each well. PCR settings were as following: 72 °C for 3 min; 98 °C for 20 s; 16 cycles of: 98 °C for 10 s, 62 °C for 1 min, 72 °C for 2 min; 72 °C 5 min; hold +4 °C. Subsequently, we pooled the contents of all the wells of a 384-well plate together and purified the resulting library using AMPure XP beads (Beckman Coulter, cat. no. A63881) at 0.8 v/v ratio. We sequenced all the libraries in paired-end mode on a NovaSeq 6000 machine using the NovaSeq 6000 SP Reagent Kit v1.5 (300 cycles) (Illumina, cat. no. 20028400). See Supplementary Data 3 for a summary of sequencing statistics.

### Computational methods

**scCircle-seq data processing.** We trimmed the sequencing reads by removing Nextera adapter sequences and overlapping R1-R2 read pairs. We then mapped the filtered reads to the human reference genome (GRCh38.p13) using the Burrows–Wheeler Aligner MEM (v0.7.17-r1188)[46] with -p flag. We removed duplicates with the *MarkDuplicates* module in Picard Tools (v2.25.5-2)[47]. We calculated the mapping rate using the *flagstat* option in SAMtools (1.13-5)[48]. To calculate the enrichment of circular over linear DNA, we divided the number of reads mapped to the circular spike-in DNA by the total number of reads mapped to all spike-in DNA (circular and linear references). If the mapping rate of a sample was less than 90% and the enrichment of circular spike-in DNA was less than 80%, we discarded the sample from downstream analyses. The success rate is above 90% for all cell lines tested in this work.

**CPR identification and classification.** To identify circle-producing regions (CPRs), we used the same approach previously described for Circle-Seq[13]. Briefly, we first identified genomic regions enriched in raw reads using the *findPeaks* option in Homer[49]. We then merged the regions in the raw circle BED file to obtain the merged circle BED file. Next, we refined the borders of the CPRs using the *closest* option in BEDtools (v2.30.0)[50] with the coverage calculated from the BAM file to get the final circle BED file. We extracted circle-supporting reads (i.e., discordant reads and split reads) from the called CPRs and filtered them with a threshold of mapQ > 20, while we removed R2R1 reads. Next, we identified chimeric junctions by extracting both ends of split reads and retained chimeric junctions with at least 2 recurrent reads within 500 base-pairs (bp) from each end of a CPR for visualization and downstream analysis. In parallel, we extracted circle-supporting reads overlapping the edge of CPRs. We calculated the circle read enrichment by dividing the number of reads mapped inside CPRs by the total number of reads.

To classify the identified CPRs, we first merged the single-cell BAM files into a pseudo-bulk BAM file for each cell type. For every CPR called in the pseudo-bulk sample, $j$, we calculated the raw frequency of occurrence for this CPR as:

$$f_{j-raw} = \frac{N_{pos}}{N_{tot}} \tag{1}$$

where $N_{pos}$ is the number of cells containing circular DNAs that overlap at least 10% of the corresponding CPRs, and $N_{tot}$ is the total number of cells profiled by scCircle-seq for the same cell line. For each single cell, $i$, we calculated the Jaccard index $J_{ij}$ between the CPR $j$ in the pseudo-bulk sample and the corresponding overlapping circles in the single cell. We then calculated the mean Jaccard index $J_{j-raw}$ by averaging the $J_{ij}$ over all the cells. Next, we normalized $f_{j-raw}$ and $J_{j-raw}$ to the frequency of occurrence of mitochondria DNA, $f_{mt}$ and the Jaccard index of mitochondria DNA, $J_{mt}$ as following:

$$f_{j-norm} = \frac{f_{j-raw}}{f_{mt}} \tag{2}$$

$$J_{j-norm} = \frac{J_{j-raw}}{J_{mt}} \tag{3}$$

and set values larger than 1 to 1. For each CPR, $j$, we calculated a uniformity score, $U_j$ as:

$$U_j = f_j \cdot J_j \tag{4}$$

We classified CPRs as high-frequency high uniformity (HFHU) when $f > 0.65$ and $U > 0.3$.

## Intersection of CPRs with enhancers

For each cell line, we downloaded the files with genomic regions containing enhancers from EnhancerAtlas 2.0[31]. We then intersected the CPRs called in single cells with the list of enhancer regions using BEDtools (v2.26.0)[50] to identify enhancer-containing circles in each cell. Next, we computed the enhancer fraction as the normalized read count from the enhancer-containing circles. Lastly, we performed motif enrichment analysis on the CPRs overlapping with enhancers using the *findMotifsGenome.pl* tool with a background file with comparable GC-content and genomic size.

## Topic modeling and dimensionality reduction

First, we filtered reads in the single-cell BAM files out using SAMtools[48] if they mapped outside the CPRs called in each single-cell BAM sample. Then, we counted the number of filtered reads in 2 kilobase (kb) genomic bins along the genome and merged the counts into a single matrix. After normalizing based on the total number of reads in each sample (single cell) and filtering out bins without any read counts, we then used the matrix as input for cisTopic[36] with default parameters for model training. We selected the best model based on the log likelihood, the second derivative of the likelihood curve, and the perplexity. Next, we subjected the topics obtained from cisTopic to dimensionality reduction and visualization using Uniform Manifold Approximation and Projection (UMAP)[42]. We annotated selected topics using the *getSignatureRegions* command in cisTopic with *minOverlap* set to 0.4. Lastly, we calculated the enrichment of genomic features across all single cells using the *AUCell_buildRankings* and *signatureCellEnrichment* commands in cisTopic. For differential topic analysis, we extracted the topic-cells matrix and used it as input for *DESeq2*[51]. Then we selected the topic pairs with adjusted $P$ value lower than 0.5 and fold-change greater than 2.

## SMART-seq2 data analysis

We first mapped the reads to the human reference transcriptome (GRCh38) using HISAT2 (v2.1.0)[52] and quantified and merged the single-cell RNA counts with RSEM[53]. We then used the merged matrix as input for Seurat (v4.0)[54] for all subsequent analyses.

## ChIP-seq data analysis

We downloaded ChIP-seq data for various histone modifications in the cell lines used in this study from the Encyclopedia of DNA Elements (ENCODE) (www.encodeproject.org) and the National Institutes of Health (NIH) Sequence Read Archive (SRA) portal (https://www.ncbi.nlm.nih.gov/sra). To calculate the enrichment of eccDNAs over specific genomic features we used the *computeMatrix* tool in deepTools (v3.5.0)[55] with the *scale-regions* flag and visualized them with the *plotProfile* tool in deepTools.

## ACT data pre-processing and copy number calling

We demultiplexed raw sequence reads to fastq files using the BaseSpace Sequence Hub cloud service of Illumina. Following this, we aligned the reads to the Hg38 reference genome using *bwa-mem* (version 0.7.17-r1188)[46]. Next, we deduplicated reads using *gatk MarkDuplicates* (version 4.2.5.0)[56]. To call absolute copy numbers in single cells we used *ASCAT.sc* (https://github.com/VanLoo-lab/ASCAT.sc). Briefly, we binned the genome in 240 kilobase (kb) bins and counted the number of reads in each bin, discarding cells with fewer than 300,000 reads. We then normalized binned read counts for GC-content using LOESS smoothing. We segmented GC-corrected read counts using the *multipcf* function from the *Copynumber* package (version 1.29.0.9)[57] with a penalty of 6. Finally, we inferred integer copy numbers using a grid search between different purity and ploidy values (purity being set to 1 due to single-cell data) and selecting the best goodness-of-fit. A small proportion of copy number profiles had extremely high and inconsistent absolute copy numbers and were filtered out by calculating the average copy number for all cells and removing cells with an average copy number >2.8.

## Statistics & reproducibility

No statistical method was used to predetermine sample size. The experiments were not randomized. The Investigators were not blinded to allocation during experiments and outcome assessment. We excluded cells yielding low-quality sequencing data from the analyses, where low quality was defined as cells for which fewer than 100 circle-producing regions (CPRs) were found and sequence data mappability was below 70%.

## Reporting summary

Further information on research design is available in the Nature Portfolio Reporting Summary linked to this article.

## Data availability

All scCircle-seq data described in this study are summarized in Supplementary Data 1. Sequencing data obtained from cell lines are publically available on the GEO database under accession code "GSE221884". Raw sequencing data from the patient samples are available on the ENA database under accession code "PRJEB71652". Pre-processed sequencing data from patient samples are available on figshare[58–60]. Previously published datasets that were used for data integration in this study are listed in Supplementary Table 1. A summary statistics of ACT data described in this study is available in Supplementary Data 3. Source data are provided with this paper.

## Code availability

All the scripts used to process and analyze the scCircle-seq data described in this study are available on github (https://github.com/BiCroLab/scCircle-seq/tree/v0.12) and zenodo[61].

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

## Acknowledgements

We thank R. P. Koche and A. R. Henssen for providing their code to call eccDNAs; X. Pan, B. Wu, and C. Chen for their help in the early stage of the project; Firas Tarish and Niklas Schultz (Karolinska Institute) for prostate sample collection and preparation; Quentin Verron (Bienko-Crosetto Lab) for FISH probe design; and Simona Pedrotti (Bienko-Crosetto Lab) for critically reading the manuscript. This work was supported by a scholarship from MIUR - Dipartimenti di Eccellenza 2018–2022 (Project No. D15D18000410001) to E.B.; by a grant from Fondazione AIRC per la Ricerca sul Cancro (grant no. 22850 under IG 2019) to C.M.; by a grant from the Swedish Cancer Society (grant no. 22 2240 Pj 01 H) to M.B.; by grants from the Swedish Cancer Society (grant no. 21 1785 Pj), the Strategic Research Programme in Cancer (StratCan) at Karolinska Institutet (grant. no. 2201), the Ragnar Soderberg Foundation (Fellows in Medicine 2016), the Swedish Research Council (grant. no. 2022-00721) and the Fondazione Piemontese per la Ricerca sul Cancro (FPRC 5xmille 2018 Ministero Salute, Programma ADVANCE-SINCE) to N.C.; and by internal core funding by Human Technopole to M.B. and N.C. Mohit Virdi is a PhD Student supported by the EURÊKA Foundation. This work was initiated in the Sunney Xie's lab in the Biomedical Pioneering Innovation Center (BIOPIC) at Peking University, whose financial support to start the project is greatly appreciated.

## Author contributions

Project conceptualization: J.P.C., C.C. Methodology: J.P.C., C.D. Sample preparation and data acquisition: J.P.C., C.D., G.D.C., E.B., K.L.G., M.V. Validation: J.P.C., H.W. Software and data analysis: J.P.C., L.H., S.E.B. Clinical knowledge support: C.M.; Funding acquisition: C.M., M.B., N.C. Project administration: J.P.C. Supervision: M.B., N.C. Visualization: J.P.C., B.A.M.B., N.C. Writing: J.P.C., B.A.M.B., M.B., and N.C. with input from all the authors.

## Funding

## Competing interests

The authors declare no competing interests.
