## [Peer review file · Nature Communications]

scCircle-seq unveils the diversity and complexity of extrachromosomal circular DNAs in single cellsREVIEWER COMMENTS

Reviewer #1 (Remarks to the Author):

The authors address the variation in circular DNA of chromosomal origin between single cells of five cancer cell lines. This is done by sequencing of 156 single cells from five cell lines. The paper describes a new method, to my knowledge the first, for isolation, sequencing and mapping of circular DNA from single cells. The authors find that there are two groups of variation in circular DNA, suggesting different levels of persistence of these circular DNAs in cell lines. One group has higher mRNA levels of oncogenes (HFHU) and appear to replicate more faithfully than the other (LFLU). Also the level of circular DNA is shown to be associated with H3K9me3 and H3K27me3.

I foresee that the work will be valuable for the growing society around circular DNA.

General comments:

The manuscript is interesting, but key questions could be addressed with the current dataset, such as what is the variation between cells in the same population, and how does the variation between two cells add up to 10E6 cells and does this correspond to the overlap found in figure 1f. If not, discuss why. And is the variation close to a random pick from the genome? The large variation shown in figure 2d, how does this fit the apparent large overlap in figure 1f?

More biologically relevant information could be given in the paper based on the current data. E.g. what are the characteristics of HFHU and LFLU, does one carry replication origins and the other one not? They appear to be oncogenes according to figure 2h, maybe emphasize this. Are there size differences between HFHU and LFLU?

Results are based on very few cells (156) and cells that are no longer under selective pressure. For instance, HeLa cells have divided trillions of times since they were isolated from Henrietta Lacks, maybe less in well-kept cell culture collections, but still many times. It would be useful for the paper if it illustrated single cell sequencing of tumor cells to get insight into cell-to-cell variation in patient samples.

The authors should adhere to the nomenclature in the field throughout the manuscript, i.e. use circular DNA, eccDNA, ecDNA or double minute as name instead of circDNA which has never been used before. The author refers to "Circular DNAs (circDNAs) have been detected in many species^{1–4} and implicated in human tumorigenesis^{5–8}". Circular DNA is denoted eccDNA, ecDNA or microDNA in these papers not

circDNA. It is so confusing for people outside the field that there are so many names for circular DNA, so there is absolutely no reason to make the situation worse by inventing a new name.

Introduction:

The authors should explain the driving hypothesis somewhat clearer in the introduction, so that it is clear why single cell sequencing is developed in this study.

Last part of the Introduction contains a repetition of the results, which are already explained in the results and the abstract. I suggest the authors rewrite to give more general ideas about the implications of their work in the last part line

68 to Line 70: The sentence only appears to apply for a limited set of eccDNA “they tend to preferentially arise from heterochromatic regions enriched in histone H3K9me3.” Other correlate with H3K4me3, which are associated with promoters. Hence this should be mentioned in line with H3K9me3 to not confuse.

Results:

Line 95: you need to explain exactly how, circle-producing regions or CPR can cover 4-16% of the genome with only 24–49 cells. This finding that each cell has 4000 - 5000 unique circles is very high and not in agreement with published Circle-Seq data, reff 9, where only a few circles were found per cell.

Line 96: rephrase accordingly. Data are not in agreement with published Circle-Seq data, reff 9.

The authors should explain how circular DNA is mapped bioinformatically in the result part.

For figure 1F it is very surprising that 49 cells carry 69% of all circles found in 10E6 cells. This must mean that most circles found in each cell is the same. A figure of the overlap between cells in each of the cell lines is needed to supplement Figure 1F.

Line 124 Of note, the high number of chimeric junctions found might be caused by the use of random primers (Exo-Resistant Random Primers mentioned in the methods). This should be mentioned. The use of “primase” instead of random primers in some Circle-Seq protocols is exactly to avoid the random effect. Mention that the apparent chimeric junctions found might be caused by the use of random primers.

Figure 2 a, b and Line 130. It is unclear what “autocorrelation of the CPR coverage as a function of genomic distance” means. What is genomic distance? Figure 2c and d are very useful and illustrative.

Line 142: HFHU regions were mainly detected in the Colo320DM cell line (8.3% of CPRs). This could be caused by many circular DNA having integrated in chromosomes of which a small number might have escaped exonuclease. Colo320DM is known to integrate the DM/circular DNA, so that consist of mixed populations of DM and hsr cells. This should be discussed/mentioned.

Line 142-149: HFHU regions were mainly detected in the Colo320DM cell line (8.3% of CPRs). This could be caused by many circular DNA having integrated in chromosomes of which a small number might have escaped exonuclease. Colo320DM is known to integrate the DM/circular DNA, so that consist of mixed populations of DM and hsr cells. This should be discussed/mentioned.

Line 151 – line 155: Findings in figure 2e are NOT in line with previous findings in Møller et al 2018, as eccDNA mapping to different genetic regions were never done in that paper. Hence, rephrase. Maybe the authors think about Shibata et al., DOI: 10.1126/science.1213307. However, The Shibata paper shows the opposite as Figure 2E (more circular DNA from 5'-regions that coding regions). This needs to be discussed in the current manuscript. I suggest the authors reanalyze the Shibata data to check if it's there is really a contradiction – if sequence data are available.

Line 160 – 163 One cannot assume that cancer cell lines and healthy human tissue (reff 9,19) have the same circular DNA patterns, and there is no contradiction in cancer cell line data being different from healthy tissue with respect to transposons. Please rephrase.

Figure 3f line 239. As far as I can see, H3K27me3 is also enriched in topic 1, 6, 12, 13, 7, 2, 4 and 9, not only. This should be mentioned in abstract and results, as its not only the mark for heterochromatin that is enriched.

Minor:

For Line 52: “Until now, three main approaches have been used to study circDNAs: (1) DNA fluorescence in situ hybridization (FISH), (2) bulk whole genome sequencing (WGS), and, more recently, (3) Circle-Seq .” The authors should refer to work methods from 60ies, 70ties and 80ties where other types of methods were used. E.g. work by G Wahl and Gaubatz.

Line 123: I think the authors should be open for enzymatic reactions being responsible for the many circDNAs of high structural complexity.

Figure 3d. There is no reference to this figure in the text. Erase or describe the figure.

Line 200 – 212: The explanation for figure 2h is not clear.

Line 528 – 529: Ethical statement and competing interests should be separate paragraphs

Reviewer #2 (Remarks to the Author):

In this work, Chen et al. developed scCircle-seq and unveils the diversity and complexity of circular DNAs in single cells. Although this is an interesting topic and such single cell technique is in need, my major criticism is that the authors did not provide proper validation for the sensitivity and accuracy of scCircle-seq. In particular, experimental approaches other than sequencing alone, eg FISH, should be implemented to validate the number and pattern of circular DNAs detected in single cells by scCircle-seq. Without knowing the false positive and false negative rates of scCircle-seq, the conclusions drawn by the authors could all be artifacts.

Point-by-point response to the Reviewers' comments

We would like to thank the Reviewers for their valuable comments and insightful suggestions. We have now revised the original manuscript to address the Reviewers' criticism, following their suggestions for improved analyses and additional experiments. In particular:

1. We have further improved the scCircle-seq protocol by adapting it for single cells or nuclei FACS-sorted in 96-well plates and reducing the original volumes to make the whole procedure more cost-efficient. We have uploaded the amended protocol to Protocol Exchange at <https://protocolexchange.researchsquare.com/article/pex-2385/v1>, as described in the revised Methods sections.
2. As suggested by Reviewer #1, we have strived to demonstrate the applicability of scCircle-seq to patient-derived tumor samples, which is essential to gain insights into the heterogeneity of eccDNAs in clinically relevant specimens. To this end, we have managed to retrieve three frozen tumor specimens (one early-stage prostate cancer, one Luminal B-like breast cancer, and one triple-negative breast cancer) from two different hospitals (Candiolo Cancer Institute, Turin, Italy and Southern Hospital, Stockholm Sweden), and processed nuclei extracted from these samples using an improved scCircle-seq workflow compatible with 96-well plates that meanwhile we developed. The results of this effort are shown in the **newly added Figure 4 and Supplementary Fig. 10-12**. Most of the eccDNAs detected in these specimens consist of low-frequency low-uniformity (LFLU) circles, recapitulating our previous findings in human cell lines. Notably, in one tumor sample (triple-negative breast cancer) that we profiled by both scCircle-seq and scDNA-seq, we uncovered two major cell subpopulations harboring distinct eccDNA repertoires. In one of these two subpopulations, eccDNAs were enriched inside regions amplified in many cells in the same tumors as well as in genomic regions frequently amplified across breast cancers in The Cancer Genome Atlas (TCGA). Based on these findings, we believe that future application of scCircle-seq to larger cohorts of tumor samples will uncover the eccDNA diversity across different tumor types, allowing in-depth analyses of the relationship between eccDNAs and linear/3D genome structure and chromatin landscape.
3. As suggested by Reviewer #2, we have designed DNA FISH probes against 10 high-frequency high-uniformity (HFHU) eccDNAs identified by scCircle-seq in PC3 and Colo320DM cells and validated the extrachromosomal nature and large cell-to-cell copy number heterogeneity of these eccDNAs by performing DNA FISH on PC3 and Colo320DM cells synchronized in metaphase. We show these new results in the **newly added Supplementary Fig. 4**. Even though the extra-chromosomal nature of eccDNAs was previously validated by DNA FISH in multiple published works (e.g., PMID: 31748743 and 34819668), our newly added data reproduce those earlier findings and further corroborate scCircle-seq results.

In sum, we believe that, thanks to the Reviewers' constructive criticism, our revised manuscript is now substantially stronger and more compelling. In particular, the demonstration of our method on patient-derived tumor samples — which was not an easy task to accomplish in the limited time frame that we were granted for revising our manuscript — represents a major improvement to the original manuscript. We therefore hope that the Reviewers will appreciate our efforts and will now support publication of our work in *Nature Communications*. Below we reply to all the Reviewers' remarks, one by one.

Reviewer #1

The authors address the variation in circular DNA of chromosomal origin between single cells of five cancer cell lines. This is done by sequencing of 156 single cells from five cell lines. The paper describes a new method, to my knowledge the first, for isolation, sequencing and mapping of circular DNA from single cells. The authors find that there are two groups of variation in circular DNA, suggesting different levels of persistence of these circular DNAs in cell lines. One group has higher mRNA levels of oncogenes (HFHU) and appear to replicate more faithfully than the other (LFLU). Also the level of circular DNA is shown to be associated with H3K9me3 and H3K27me3.

I foresee that the work will be valuable for the growing society around circular DNA.

The manuscript is interesting, but key questions could be addressed with the current dataset, such as what is the variation between cells in the same population, and how does the variation between two cells add up to 10E6 cells and does this correspond to the overlap found in figure 1f. If not, discuss why. And is the variation close to a random pick from the genome? The large variation shown in figure 2d, how does this fit the apparent large overlap in figure 1f?

We thank the Reviewer for appreciating our work and for their time and constructive comments and suggestions, which helped us deepen our analyses and present our data in a clearer and more compelling way.

Regarding the Reviewer's question on the eccDNA variability, we have re-done the comparison between bulk Circle-Seq and scCircle-seq, by first creating an ensemble of all the circle-producing regions (CPRs) detected by scCircle-seq across multiple cells and then comparing this 'pseudo-bulk' dataset with the CPR pattern detected by Circle-Seq in the same cell line (Colo320DM). Except for high-frequency eccDNAs that show a good correspondence with scCircle-Seq, most of the regions identified by bulk Circle-Seq only provide an average picture of multiple eccDNAs originating in different single cells, often resulting in flattening the profile out and hiding individual events coming from single cells. Instead, in the same regions scCircle-Seq is able to detect multiple individual eccDNAs that largely vary (in their position within the same region) from one cell to another, which we name low frequency low uniformity (LFLU) eccDNAs (see **new Supplementary Fig. 2a, b**). What we find in pseudo-bulk scCircle-seq data is that eccDNAs can generally arise anywhere along the genome, however there are clear preferences for gene bodies and chromatin marked by H3K9me3 and H3K27me3 (see **Fig. 2e, f**). Therefore, although the cell-to-cell variability in scCircle-seq data is very high, the distribution of CPRs in ensemble scCircle-seq data is not random, even in the case of LFLU eccDNAs. This explains the relatively good correspondence between bulk Circle-Seq and (pseudo-bulk) scCircle-seq data shown in the Venn diagram in the original Fig. 1f. To avoid confusion, in the revised manuscript we no longer show this plot but instead use the **new Supplementary Fig. 2** and revised main text to convey the above considerations more clearly.

More biological relevant information could be given in the paper based on the current data. E.g. what are the characteristics of HFHU and LFLU, does one carry replication origins and the other one not? They appear to be oncogenes according to fig 2h, maybe emphasize this. Are there size differences between HFHU and LFLU?

We thank the Reviewer for these relevant questions and suggestions for further analyses. We previously examined whether HFHU and LFLU CPRs are enriched in different replication timing regions, by leveraging publically available OK-Seq or Repli-Seq data from

ENCODE. However, we did not observe a clear difference of replication timing between the two types of regions. We also note that, unfortunately, neither OK-Seq or Repli-Seq data are available in ENCODE (and, to our knowledge, in none of the other publically available repositories) for the two cell lines (Colo320DM and PC3) that we find carrying the highest amount of HFHU CPRs. Therefore, we cannot conclusively exclude that a subset of highly abundant eccDNAs is related to replication timing.

Regarding oncogenes, indeed they are enriched in HFHU eccDNAs and we have now emphasized this in the revised manuscript. We thank the Reviewer for this suggestion.

Regarding the difference in size between HFHU and LFLU eccDNAs, we need to distinguish between the size of the regions from which they arise (which we name circle-producing regions or CPRs) and the length of eccDNAs themselves. LFLU CPRs are larger than HFHU CPRs at the ensemble level but give rise to multiple smaller eccDNAs whereas each HFHU CPR typically forms a single, large eccDNA molecule (which in the literature is named ecDNA). Therefore, while LFLU CPRs are larger, the eccDNAs originating from them are typically smaller, as we now show in the newly added **Supplementary Fig. 3b and c**. We apologize if this difference was not clearly explained in our original manuscript and have now tried to improve this explanation in the revised manuscript.

It would be useful for the paper if it illustrated single cell sequencing of tumor cells to get insight into cell-to-cell variation in patient samples.

We thank the Reviewer for this excellent suggestion. Accordingly, in the past few months we have managed to retrieve three frozen tumor specimens (one early-stage prostate cancer, one Luminal B-like breast cancer, and one triple-negative breast cancer) from two different hospitals (Candiolo Cancer Institute, Turin, Italy and Southern Hospital, Stockholm Sweden). This was no trivial task, especially considering the administrative and logistic difficulties associated with retrieving samples from two different hospitals during the summer holidays period. The results of this effort are shown in the **newly added Figure 4 and Supplementary Fig. 10-12**. We processed nuclei extracted from these samples using an improved scCircle-seq workflow compatible with 96-well plates that meanwhile we have developed and that we now make fully available at Protocol Exchange (<https://protocolexchange.researchsquare.com/article/pex-2385/v1>). By performing UMAP and topic analysis on these new datasets, we found that single-cell eccDNA profiles can clearly distinguish between prostate and breast cancer tumors, reflecting their different tissue of origin. Moreover, by focusing our analysis on the two breast cancer samples, we uncovered two clearly distinct clusters of cells from the triple-negative breast cancer (TNBC), with cells from the Luminal B-like tumor 'bridging' the two clusters. Comparison of eccDNA profiles with genome-wide DNA copy number profiles of breast cancers in TCGA showed that TNBC Cluster-1 eccDNAs are significantly enriched in amplified regions, suggesting that more eccDNAs are produced from genomic regions in excess number of copies. To gain further insights into this possibility, we performed single-cell DNA sequencing (using a method, ACT, previously developed to investigate the heterogeneity of breast cancers [PMID: 33762732]) on nuclei extracted from the same TNBC sample to call amplifications (AMP) and deletions (DEL) in the same tumor sample profiled by scCircle-seq. In line with the results of our comparison with TCGA data, we found that the eccDNAs identified in Cluster-1 are indeed enriched in AMP regions detected in the same sample. Notably, we did not uncover a linear correlation between DNA copy number and eccDNA abundance (only highly amplified regions were associated with a significantly higher amount of eccDNAs), indicating that the relationship between genomic alterations

and eccDNA formation is more complex. Based on these proof-of-principle findings, we believe that future application of scCircle-seq to larger cohorts of tumor samples will allow uncovering the full spectrum of eccDNA heterogeneity across different tumor types and enable in-depth analyses of the relationship between ecDNAs and linear/3D genome structure and chromatin landscape. However, we hope that the Reviewer will agree that extending our analysis to more tumor samples would not be feasible in the context of this revision and should be part of a future follow-up study.

The authors should adhere to the nomenclature in the field throughout the manuscript, i.e. use circular DNA, eccDNA, ecDNA or double minute as name instead of circDNA which has never been used before.

We thank the Reviewer for the suggestion. Accordingly, we now use the term eccDNAs throughout the manuscript except when referring to large circular DNAs harboring oncogenes, for which we use the term ecDNA instead, following the prevalent nomenclature in the field (even though, a clear consensus does not exist yet).

The authors should explain the driving hypothesis somewhat clearer in the introduction, so that it is clear why single cell sequencing is developed in this study.

We have now tried to better motivate in the Introduction why we need methods for single-cell profiling of eccDNAs, emphasizing the need for innovative single-cell sequencing approaches to characterize the full spectrum of genetic and epigenetic intra-tumor heterogeneity across different tumor types. We hope that the Reviewer will find our motivation for developing scCircle-seq clearer and more convincing.

Circle-producing regions or CPR can cover 4-16% of the genome with only 24–49 cells. This finding that each cell has 4000 - 5000 unique circles is very high and not in agreement with published Circle-Seq data, reff 9, where only a few circles were found per cell.

We thank the Reviewer for this important consideration. In the published bulk Circle-Seq data, the number of detected eccDNAs range from ~600 (normal blood) to ~20,000 (muscle from normal sample) with 40 to 200 million read pairs (2x50 nt) sequenced. In contrast, in this study we have generated ~10 million read pairs (2x150 nt) for every single cell — a rather high sequencing depth that should enable capturing a large repertoire of eccDNAs in each cell. Importantly, by using milder lysis conditions compared to the original Circle-Seq protocol and introducing a nick repair step to minimize the formation of artefactual nicks in the eccDNA (which would cause the circular DNA to be digested by Exo V), we have managed to increase the number of eccDNAs detected from ~500 per cell on average to 3,000 on average (in PC3 cells) (see **Supplementary Fig. 1e-h**). We also note that the majority of eccDNAs detected by scCircle-seq are present only in a fraction of the cells sequenced. Therefore, in bulk Circle-Seq these eccDNAs might be missed, explaining the lower number of eccDNAs detected in comparison to scCircle-seq.

Another possible explanation of the discrepancy mentioned by the Reviewer lies in the different algorithms used to call eccDNAs in bulk Circle-Seq vs. scCircle-seq. In Circle-Seq (PMID: 29540679; 31989524), eccDNAs are detected using an algorithm (Circle-map) that relies on split and discordant reads to find breakpoints. This method is very effective to identify highly overrepresented eccDNAs with high precision (even at base-pair resolution). However, when we tried to apply Circle-map to scCircle-seq data, we found that the algorithm often fails to identify breakpoints when the reads around them are of low mapping quality, thus resulting in a significantly lower number (on average, 800 per cell) of eccDNAs

identified compared to using our pipeline. Hence, different algorithms can result in different numbers of eccDNAs identified. We decided to use our algorithm because, unlike Circle-map, it can filter out most of the non-enriched reads (likely coming from residual linear gDNA) leaving only bona fide eccDNA reads for downstream analysis. Having said that, we acknowledge that, especially when the coverage per cell is low, genomic regions corresponding to a single eccDNA might be artificially split into half, contributing to the higher number of eccDNAs identified by scCircle-seq compared to bulk Circle-Seq.

Lastly, we would like to rectify an imprecision that might have confused the Reviewer: 4–16% coverage, as reported in our original manuscript, referred to the total number of reads in each single cell and not to the true CPR coverage. Since only 40–70% of the reads are bona fide eccDNA reads in our dataset (i.e., 30–60% of the reads are discarded by our pipeline as they are not enriched in specific regions), the true coverage of CPRs is 1.5–8%. We have now clarified this in our revised manuscript.

The authors should explain how circular DNA is mapped bioinformatically in the result part.

We thank the Reviewer for pointing this out. Our bioinformatic pipeline to detect eccDNAs is based on peak calling and sequencing coverage and therefore indirectly detects eccDNAs by identifying so-called circle producing regions or CPRs. Briefly, we first identify high-coverage genomic regions enriched in aligned read pairs and, in parallel, call read peaks with variable sizes (typically, high-coverage regions are broader than peaks). We then overlap high-coverage read blocks with peaks to call CPRs and refine their edges by finding the closest edges of enriched read blocks to the peaks. Lastly, after calling CPRs, we identify chimeric junctions by finding junctions with both ends supported by recurrent high-mapQ chimeric reads (> 3 reads within 500 bp). We have now added a short version of this description in the Results section, while we explain the procedure in more details in the Methods section.

For figure 1F it is very surprising that 49 cells carry 69% of all circles found in 10E6 cells. This must mean that most circles found in each cell is the same. A figure of the overlap between cells in each of the cell lines is needed to supplement Figure 1F.

As already discussed above, different cells have different sets of LFLU eccDNAs and the overlap between the corresponding CPRs across different cells is very low. However, despite the high heterogeneity of LFLU eccDNAs, the genomic regions in which these eccDNAs arise are typically large thus explaining the overlap with CPRs called based on bulk Circle-Seq data. To avoid confusion, we have now removed the Venn diagram from Figure 1 and instead compare scCircle-seq with Circle-Seq in the **new Supplementary Fig. 2**.

Line 124 Of note, the high number of chimeric junctions found might be caused by the use of random primers (Exo-Resistant Random Primers mentioned in the methods). This should be mentioned. The use of “primase” instead of random primers in some Circle-Seq protocols is exactly to avoid the random effect. Mention that the apparent chimeric junctions found might be caused by the use of random primers.

The use of high concentration of random primers could certainly contribute to the formation of split and discordant reads during the RCA step in scCircle-seq. To minimize this risk experimentally, we limit the time of RCA to 3 hours to prevent artefacts caused by displacement failure and overamplification. Moreover, at the computational level, we call

chimeric junctions from genomic ends with recurrent deduplicated split read pairs. Using such approach, we were able to identify only a single head-to-tail chimeric junction using a spike-in plasmid sample amplified with RCA, as shown in **Supplementary Fig. 1c**. Furthermore, since the random generation of split and discordant reads results from displacement failure of the Phi29 enzyme in the presence of high concentrations of random primers, we leveraged MALBAC, which uses a low random primer concentration and multiple rounds of denaturation, to assess whether RCA is a major source of chimeric read artefacts in scCircle-seq. As shown in **Supplementary Fig. 1i-k**, the number of chimeric junctions called for mtDNA are comparable between RCA and MALBAC, indicating that the relatively high number of chimeric junctions identified by scCircle-seq is unlikely an artefact caused by the use of RCA.

Figure 2 a, b and Line 130. It is unclear what “autocorrelation of the CPR coverage as a function of genomic distance” means. What is genomic distance? Figure 2c and d are very useful and illustrative.

We thank the Reviewer for appreciating our visual classification of different classes of eccDNAs. We have now added an additional scheme in the **new Fig. 2a** to explain how we compute autocorrelation. Generally speaking, autocorrelation refers to the correlation between a signal and a distance or time-delayed version of itself. In our specific application, autocorrelation refers to the correlation between the scCircle-seq signal $f(x)$ (i.e., the genome-wide distribution of sequencing reads coming from eccDNAs) and the same signal shifted of a genomic distance, d ($f(x+d)$ in **Fig. 2a**). For single cells, the correlation drops very quickly as a function of increasing genomic distance, which means that the formation of an eccDNA at a specific genomic location is independent of the formation of another eccDNA at another location, even when the distance between the two locations is very small. However, for merged (pseudo-bulk) scCircle-seq datasets, the autocorrelation drops much slower at increasing genomic distances, indicating that eccDNAs tend to form more frequently than by chance in some genomic regions. We have now added a short version of this explanation in the corresponding Results part, hoping to make it clearer for the Readers how to interpret the plots shown in **Fig. 2b and c**.

Line 142/ Line 142-149: HFHU regions were mainly detected in the Colo320DM cell line (8.3% of CPRs). This could be caused by many circular DNA having integrated in chromosomes of which a small number might have escaped exonuclease. Colo320DM is known to integrate the DM/circular DNA, so that consist of mixed populations of DM and hsr cells. This should be discussed/mentioned.

We thank the Reviewer for this comment and suggestion. Following the remarks of Reviewer #2, we have performed DNA FISH to validate some of the HFHU eccDNAs detected by scCircle-seq in three different cell lines and indeed observed that some of these eccDNAs are possibly integrated back into chromosomal DNA in Colo320DM cells. The results of these experiments are shown in the **new Supplementary Fig. 4**. We hope that the Reviewer will appreciate our effort to validate scCircle-seq using an orthogonal method.

Line 151 – line 155: Findings in figure 2e are NOT in line with previous findings in Møller et al 2018, as eccDNA mapping to different genetic regions were never done in that paper. Hence, rephrase. Maybe the authors think about Shibata et al., DOI: 10.1126/science.1213307. However, The Shibata paper shows the opposite as Figure 2E (more circular DNA from 5'-regions that coding regions). This needs to be discussed in the current manuscript. I suggest

the authors reanalyze the Shibata data to check if it's there is really a contradiction – if sequence data are available.

We are grateful to the Reviewer for the suggestions and apologize for the imprecision, which we have rectified in the revised manuscript. We have now downloaded Circle-Seq data from Shibata et al 2012, Møller et al 2018, and Yuangao et al 2021 and re-analyzed them to assess whether a depletion of eccDNAs around the TSS can also be detected in those datasets. As shown in the **newly added Supplementary Fig. 5**, a signal depletion around the TSS can be seen in both the Møller and Yuangao datasets, whereas it cannot be appreciated in the Shibata data. However, we note that the latter paper focused on microDNAs (0–1 kb), whereas a wider repertoire of eccDNAs were analyzed in Møller et al 2018 (0.1–10 kb), Yuangao et al 2021 (0.4–3 kb) and our work (0.1 kb–1 Mb). Interestingly, in our topic modelling analysis, although most of the cell-type specific topics are depleted in 5' UTRs (topics 3, 6, 7, 11, 14), we find that some background topics (1,2,9) are enriched in 5' and 3' UTRs, indicating that a subset of non-cell-type specific eccDNAs form in these regions, matching the enrichment analysis in Shibata et al 2012 (see **Fig. 2a** in that paper). In conclusion, we believe that our results are not contradicting previous studies and that the differences observed are likely attributable to the different repertoires of eccDNAs analyzed in different studies.

Line 160 – 163 One cannot assume that cancer cell lines and healthy human tissue (reff 9,19) have the same circular DNA patterns, and there is no contradiction in cancer cell line data being different from healthy tissue with respect to transposons. Please rephrase.

We thank the Reviewer for pointing this out and have rephrased the statement accordingly.

Figure 3f line 239. As far as I can see, H3K27me3 is also enriched in topic 1, 6, 12, 13, 7, 2, 4 and 9, not only. This should be mentioned in abstract and results, as its not only the mark for heterochromatin that is enriched.

In the original manuscript, we only mentioned cell type-specific topics (3, 6, 7, 11, 14) that were enriched in H3K9me3 from the corresponding cell line. Topics 1, 2, 4, 9, 12, and 13 were not cell type specific and therefore we did not include them. We admit that some cell type specific topics (e.g., topic 3 and 7) are indeed somehow enriched in H3K27me3, but the enrichment is not as pronounced as for H3K9me3, as shown in **Fig. 3f**. However, we have now added H3K27me3 in the Abstract to provide the Readers with a summary more adherent to our findings.

For Line 52: “Until now, three main approaches have been used to study circDNAs: (1) DNA fluorescence in situ hybridization (FISH), (2) bulk whole genome sequencing (WGS), and, more recently, (3) Circle-Seq .” The authors should refer to work methods from 60ies, 70ties and 80ties where other types of methods were used. E.g. work by G Wahl and Gaubatz.

We thank the Reviewer for this remark and apologize for having omitted citing those earlier studies. We have now added the suggested reference in the revised Introduction.

Line 123: I think the authors should be open for enzymatic reactions being responsible for the many circDNAs of high structural complexity.

We thank the Reviewers for this comment. As discussed above, we use different strategies to limit the possibility that artefact circular DNAs are formed during the scCircle-seq

workflow. However, we cannot completely rule out that some of the 'complex' eccDNA identified by scCircle-seq represent false-positive events that arise during the various enzymatic reactions included in the scCircle-seq protocol, especially during the RCA step. We have now added these considerations in the corresponding revised part in the Results section.

Figure 3d. There is no reference to this figure in the text. Erase or describe the figure.

We apologize for this and have now added the missing reference in the revised manuscript.

Line 200 – 212: The explanation for figure 2h is not clear.

We have now rephrased this part and hope that the Reviewer will find it clearer.

Line 528 – 529: Ethical statement and competing interests should be separate paragraphs

Following the Journal's guidelines, we have now added an 'Inclusion & Ethics statement' in which we mention the ethical permits related to the use of clinical samples. In the 'Competing interests' section we state that we do not have any relevant conflict of interest.

Reviewer #2

In this work, Chen et al. developed scCircle-seq and unveils the diversity and complexity of circular DNAs in single cells. Although this is an interesting topic and such single cell technique is in need, my major criticism is that the authors did not provide proper validation for the sensitivity and accuracy of scCircle-seq. In particular, experimental approaches other than sequencing alone, eg FISH, should be implemented to validate the number and pattern of circular DNAs detected in single cells by scCircle-seq. Without knowing the false positive and false negative rates of scCircle-seq, the conclusions drawn by the authors could all be artifacts.

We thank the Reviewer for acknowledging the need to develop methods to study eccDNAs in single cells as well as for their suggestion to validate some of the eccDNAs detected by scCircle-seq with an orthogonal approach. Accordingly, we have now leveraged the iFISH pipeline for high-resolution DNA FISH, which we previously established (PMID: 30967549), to visualize 10 different high frequency high uniformity (HFHU) eccDNAs identified by scCircle-seq in Colo320DM cells and 3 HFHU eccDNAs identified in PC3 cells. As shown in the **new Supplementary Fig. 4**, by performing iFISH on Colo320DM and PC3 cells synchronized in metaphase, we were able to detect many signals outside of metaphase chromosomes (and not inside interphase nuclei), confirming the extrachromosomal nature of these DNA fragments and further validating our method. In PC3 cells, the number of eccDNA signals detected was considerably inferior compared to Colo320DM cells. This is in line with previous studies based on bulk sequencing and cytogenetic data, which reported a low eccDNA copy number in PC3 cells (PMID: 28178237; 31748743). We note that the same HFHU eccDNAs that we visualized in Colo320DM cells were also previously validated by DNA FISH in the same cell line (PMID: 31748743, 34819668), further corroborating our findings.

Regarding scCircle-seq false-positive and -negative rates, our experiment performed using a mixture of plasmid and linear DNA yielded over 97% of the reads from plasmid DNA, and hence a false-positive rate lower than 3%, as shown in **Supplementary Fig. 1b**. Moreover, in all our scCircle-seq experiments we spiked equivalent amounts of plasmid and linear gDNA into the lysis buffer, always obtaining a high enrichment (typically over 1,000) of plasmid over linear DNA, demonstrating the ability of our approach to enrich for circular DNA. This is further highlighted by the fact that, in most (248 out of 250) of the single cells that we have profiled by scCircle-seq, we have consistently been able to detect mitochondrial DNA (mtDNA) (which is circular) and the cells that have low levels of mtDNA are low-quality cells filtered out from our analysis.

We hope that the Reviewer will find that our revised manuscript — which contains new data as well as more in-depth analyses and, in particular, a proof-of-principle application of scCircle-seq to tumor samples — more convincingly demonstrates the technical validity and utility of scCircle-seq, and therefore will support its publication in *Nature Communications*.

REVIEWER COMMENTS

Reviewer #1 (Remarks to the Author):

I have no more comments to the manuscript since all my initial comments have been addressed in this second version of the manuscript.

Reviewer #3 (Remarks to the Author):

I was specifically asked to review the manuscript and focus on comments made by reviewer 2 with regards to validation of the eccDNAs identified using their new single cell method.

The new data falls short of providing this important validation in my view. The reviewer reasonably asked for an orthogonal method to validate false positive and negative rates.

The vast majority of eccDNAs detected by scCircle-seq technique are LFLU. These were not validated. The 10 HFHU eccDNAs selected for validation by FISH would presumably also be detected by bulk methods so the FISH validation does not in any way validate the veracity of the single cell data.

In my view what is required here is to examine some regions where eccDNA was not detected and show background FISH signal outside of metaphase chromosomes. The authors should then show some LFLU and HFHU ecc DNAs and show the correlation of the expected inter-cell heterogeneity from their scCircle-seq with heterogeneity identified by FISH. This will then validate that the heterogeneity they observe is real biology rather than technical noise, which is the whole purpose of their assay.

Point-by-point response to the Reviewers' comments

We would like to thank both Reviewers for their valuable time and constructive comments. We are pleased that the original Reviewer #1 is satisfied by our revisions. Regarding the remarks of Reviewer #3 substituting previous Reviewer #2, while we understand their wish for additional validations by DNA FISH, we believe the experiment proposed by the Reviewer would not be informative, as we try to explain below.

We therefore sincerely hope that also Reviewer #3 will be open to recognize that our previous revisions — including orthogonal validation of high-frequency eccDNAs by DNA FISH — provide sufficient evidence of the technical reliability and applicability of our method to both biologically and clinically relevant samples, thus warranting publication of our work in *Nature Communications*.

Reviewer #1

I have no more comments to the manuscript since all my initial comments have been addressed in this second version of the manuscript.

We are grateful to the Reviewer for appreciating our effort to further strengthen our manuscript, making it suitable for the broad readership of *Nature Communications*.

Reviewer #3

I was specifically asked to review the manuscript and focus on comments made by reviewer 2 with regards to validation of the eccDNAs identified using their new single cell method.

The new data falls short of providing this important validation in my view. The reviewer reasonably asked for an orthogonal method to validate false positive and negative rates.

The vast majority of eccDNAs detected by scCircle-seq technique are LFLU. These were not validated. The 10 HFHU eccDNAs selected for validation by FISH would presumably also be detected by bulk methods so the FISH validation does not in any way validate the veracity of the single cell data.

In my view what is required here is to examine some regions where eccDNA was not detected and show background FISH signal outside of metaphase chromosomes. The authors should then show some LFLU and HFHU ecc DNAs and show the correlation of the expected inter-cell heterogeneity from their scCircle-seq with heterogeneity identified by FISH. This will then validate that the heterogeneity they observe is real biology rather than technical noise, which is the whole purpose of their assay.

We thank the Reviewer for critically assessing our response to the previous comments by Reviewer #2. Although we agree with the Reviewer that it would be valuable to validate the LFLU eccDNAs detected by scCircle-seq, unfortunately **this is not possible because LFLU eccDNAs are stochastic events** that are randomly generated and therefore are individually detectable only in one cell. It is true that LFLU circles tend to preferentially originate from certain genomic regions (as we amply demonstrate in our manuscript), however individual LFLU eccDNAs are essentially random molecular events that cannot be detected across multiple cells. Therefore, it is not possible to validate an LFLU eccDNA

detected by scCircle-seq in one cell by performing DNA FISH on another cell using a probe targeting that exact LFLU circle. (It remains possible that, if a very large number of single cells were to be profiled, some LFLU eccDNAs would be identified in more than one cell. However, to test this possibility one would need to sequence a very large number of cells (which we believe goes beyond the budget limits of an individual research group).

Based on these considerations, in our previous revision we reasoned that the best way to address the request of Reviewer #2 << to validate the number and pattern of circular DNAs detected in single cells by scCircle-seq >> would be to validate by DNA FISH some of the high-frequency high-uniformity (HFHU) eccDNAs detected by scCircle-seq across multiple cells. This was possible because, unlike LFLU eccDNAs, HFHU circles are non-random events detected in multiple cells. Indeed, using DNA FISH, we were able to validate several of the HFHU eccDNAs detected by scCircle-seq, further demonstrating the reliability of our method. However, even in this experimental setting, we could not directly compare the frequency of HFHU eccDNAs detected by sequencing vs. DNA FISH, because DNA FISH needs to be performed on cells synchronized in metaphase, and only a fraction of the cell population can be synchronized in metaphase at any given time (specifically, in the case of Colo520DM cells that we used, we did not manage to get a high percentage of cells synchronized in metaphase, despite numerous attempts).

Of note, our observation that most of the eccDNAs captured by scCircle-seq are LFLU is recapitulated in a paper published in *Nature Genetics* earlier this year (<https://www.nature.com/articles/s41588-023-01386-y>), which describes another single-cell eccDNA detection method (scEC&T-seq). Notably, the Authors of that paper were not able to validate the low-frequency random eccDNAs identified with their method but instead could only validate high-frequency eccDNAs by FISH — exactly as we have done in our previous revision.

We would also like to point out that, in the single-cell omics field, there are other examples of unrepeatable stochastic molecular events detected in single cells that cannot be validated. For example, the heterogeneous inter-chromosomal contacts detected by scHi-C assays cannot be validated by DNA FISH simply because those contacts occur randomly, and the same contact cannot be detected across multiple cells.

If the Reviewer thinks it's a good idea, we are happy to include these considerations in the Discussion section.

In sum, we sincerely hope that the Reviewer will agree with us that the additional validation experiment proposed is not feasible and you will be open to accept our current manuscript for publication in *Nature Communications* without this additional experiment.